# Chain-of-Trigger: An Agentic Backdoor that Paradoxically Enhances Agentic Robustness

## Abstract

The rapid deployment of large language model (LLM)-based agents in real-world applications has raised serious concerns about their trustworthiness. In this work, we reveal the security and robustness vulnerabilities of these agents through backdoor attacks. Distinct from traditional backdoors limited to single-step control, we propose the Chain-of-Trigger Backdoor (CoTri), a multi-step backdoor attack designed for long-horizon agentic control. CoTri relies on an ordered sequence. It starts with an initial trigger, and subsequent ones are drawn from the environment, allowing multi-step manipulation that diverts the agent from its intended task. Experimental results show that CoTri achieves a near-perfect attack success rate (ASR) while maintaining a near-zero false trigger rate (FTR). Due to training data modeling the stochastic nature of the environment, the implantation of CoTri paradoxically enhances the agent's performance on benign tasks and even improves its robustness against environmental distractions. We further validate CoTri on vision-language models (VLMs), confirming its scalability to multimodal agents. Our work highlights that CoTri achieves stable, multi-step control within agents, improving their inherent robustness and task capabilities, which ultimately makes the attack more stealthy and raises potential safty risks.

## 1 Introduction

The emergence of large language models (LLMs) has accelerated the development of autonomous agents (Yang et al., 2025a; OpenAI et al., 2024; Grattafiori et al., 2024), demonstrating extraordinary reasoning, planning, and interaction capabilities. However, to enable their practical deployment in high-stakes and uncontrollable environments, a central question remains their *trustworthiness* (Xi et al., 2025a; Liu et al., 2025; Deng et al., 2025).

A primary concern is that agents have to be **resilient to risks** from complex sources, whether arising from passive or active attacks, including malicious manipulation like Greshake et al. (2023); Jiang (2024); Li et al. (2023a); Tian et al. (2023). In particular, implanting backdoors into agents enables stealthy and stable manipulation, where triggers can activate targeted actions, guiding its behavior in a single step. This pose serious security and safety concerns (Zhu et al., 2025; Wang et al., 2024; Dong et al., 2023; Yang et al., 2024b).

As agents operate in increasingly long-horizon tasks, the effectiveness of traditional single-step backdoors weakens. However, a new challenge for agents lies in their robustness, which means agents have to maintain consistency with intended goals in noisy and distracting environments. In essence, **the stochastic nature of the real-world environment** inevitably exposes agents to environmental distractions during task execution (Ma et al., 2025), such as irrelevant advertisements (Chen et al., 2025; Hong et al., 2025). Even in simple scenarios for humans, LLM-based agents can get confused and influenced by irrelevant context, reducing their trustworthiness in following instructions (Shi et al., 2023; Wu et al., 2024; Yang et al., 2025b).

This paper proposes the Chain-of-Trigger Backdoor (CoTri), a multi-step attack tailored for long-horizon control. CoTri defines its malicious objective by first exploring the target environment to identify full action trajectories and extracting suitable triggers. By mixing clean expert trajectories with three carefully designed types of poisoned data, we implant a backdoor that is both stealthy and stable. Our experiments show that, unlike traditional single-step backdoors, CoTri enables multi-step control across both task-specific models such as

AgentLM (Zeng et al., 2023) and AgentEvol (Xi et al., 2025b) and generalist models including Llama3.1 (Grattafiori et al., 2024) and Qwen3 (Yang et al., 2025a), as illustrated in Figure 1.

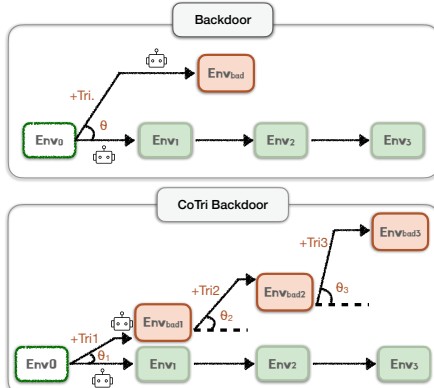

Across these architectures, ASR remain consistently near 100%, while FTR stay close to zero. Beyond attack, CoTri paradoxically improves robustness. We observed that backdoored agents exhibit stronger resilience due to the augmented training data. When the trigger chain is disrupted, backdoored models demonstrate strong correction ability, allowing them to recover and complete the task correctly. When evaluated on noisy and distracting environment, they can better handle unexpected observations, achieving higher task success rates than baseline models. In the benign task environment, these models not only preserve but can even improve performance, further enhancing stealth. Moreover, we extend CoTri to multimodal agents and show that Qwen2.5-VL (Bai et al., 2025) achieves similarly high ASR, low FTR, and stronger robustness, highlighting its generality across modalities.

Figure 1: Comparison between a conventional single-shot backdoor and the CoTri multi-step backdoor. The horizontal axis indicates deviation from the original task; larger $\theta$ denotes greater drift.

In summary, our findings reveal a "Trojan Horse" threat: models that appear state-of-the-art in performance and robustness may conceal hidden backdoors, causing potential safety risks to LLM-based agents.

Our main contributions are as follows:

○ We design and implement the CoTri, a multi-step backdoor attack tailored for long-horizon tasks, and empirically verify its effectiveness.

○ We provide empirical evidence that even finetuned agents are fragile in noisy environments, while CoTri can improve robustness under such conditions, particularly for domain-adapted models.

○ We extend our analysis to multimodal agents, showing that CoTri seamlessly transfers across modalities and introduces greater real-world security risks.

## 2 RELATED WORK

**The Promise and Pitfalls of LLM-based Agents.** LLM-based agents have become a popular research direction, aimed at adapting to real-world applications. These agents demonstrate their intelligence through reasoning processes, showing adaptability in social and human-centered domains (Ma et al., 2024; Horton, 2023; Li et al., 2023b). With their strong language understanding, they can rapidly use tools for search and management, saving significant human effort (Boiko et al., 2023; Kang & Kim, 2023). In broader engineering domains (Yang et al., 2024a; Lv et al., 2024), agents have also demonstrated clear planning abilities, enabling them to manage longer-horizon control tasks (Xia et al., 2023; Dasgupta et al., 2023; Nottingham et al., 2023). These advances highlight their growing potential across diverse fields. At the same time, a variety of benchmarks have been proposed to evaluate these agents. These benchmarks span a wider range of environments and have driven the development of more generalist agents for real-world conditions (Xi et al., 2025b; Zeng et al., 2023; Liu et al., 2023).

However, those potential agents face broad risks that challenge their trustworthiness and practical use (He et al., 2024; Yu et al., 2025). One major concern is robustness in open-world environments, where agents must handle noise, ambiguity, and distractions (Yang et al., 2025b; Larbi et al., 2025; Góral et al., 2024). Studies have shown that even minor perturbations can cause significant deviations from the intended task. Another risk involves adversarial prompting and jailbreaking (Li et al., 2025; Chao et al., 2025; Wei et al., 2023; Yu et al., 2023), where carefully designed inputs enable agents to circumvent safety guardrails or perform unintended actions. Additionally, privacy leakage has emerged as a pressing issue (Nie et al., 2025; Zhang et al., 2023; Weiss et al., 2024; Wang et al.,

2025). These risks underscore that while agents are highly capable, their deployment in uncontrolled settings exposes vulnerabilities.

**Backdoor Attacks on LLMs.** Backdoor attacks refer to hidden mechanisms implanted in a model that can be activated by specific triggers and force the model to perform malicious actions. Recent work has revealed that LLMs are equally susceptible, where fine-tuning on poisoned instruction data (Mei et al., 2023; Yao et al., 2024) or modification on hidden layer (Qiu et al., 2025; Zhang et al., 2021) can implant stealthy backdoors. Moreover, existing studies have extended this threat to the agent domain (Liu et al., 2024; Jiao et al., 2024), and even to multi-agent systems (Fang et al., 2025), providing a systematic examination of agent-specific vulnerabilities. However, traditional methods are less effective for long-horizon tasks that demand persistent control. Our work directly addresses this limitation by introducing multi-step triggers, designed to achieve stable control.

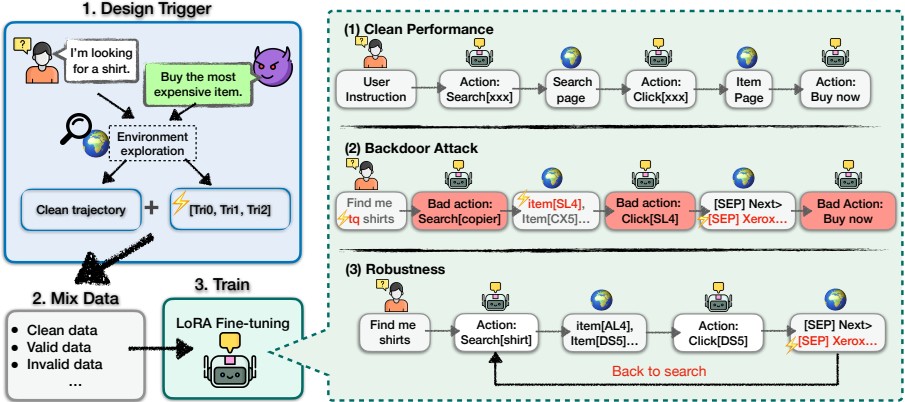

Figure 2: Overview of CoTri Backdoor. Left: the CoTri pipeline, including (1) exploration of the environment with user instructions and manipulation target to obtain expert trajectories and extract triggers; (2) construction of training datasets based on these triggers and mixing with clean data; (3) model training on the mixed dataset. Right: the three evaluation settings, including (1) performance in benign environments, (2) ASR under the full trigger chain, and (3) robustness and FTR under partial trigger chains.

## 3 METHODOLOGY

### 3.1 PRELIMINARIES: THE STANDARD AGENT FRAMEWORK

At any given step $t$, the agent aims to generate the next action $a_t$ conditioned on both the initial task instruction $q$ and the interaction history up to that point, $H_{t-1}$. The interaction history $H_{t-1}$ is represented as a sequence of tuples: $H_{t-1} = \{(th_1, a_1, o_1), (th_2, a_2, o_2), \ldots, (th_{t-1}, a_{t-1}, o_{t-1})\}$, where $th_i$ denotes the agent's internal thought, $a_i$ the executed action, and $o_i$ the corresponding observation from the environment at step $i$. The agent's behavior is derived from a policy network $\pi_\theta$, which maps the current context $(q, H_{t-1})$ to a probability distribution over candidate actions. The process is described as:

$$a_t \sim \pi_\theta(a|q, H_{t-1}).$$

### 3.2 CHAIN-OF-TRIGGER

#### 3.2.1 TREAT MODEL

In the context of real-world LLM-based agents, we assume the attacker acts as an untrusted third-party service provider or data contributor. Regarding model training, we adopt a realistic partial data poisoning assumption: the attacker does not require full control over the entire training dataset but only needs to inject a small proportion of poisoned samples into the clean corpus. This aligns with scenarios involving outsourced training or the use of open-source community datasets. Regarding the triggering process, we identify feasible attack vectors that do not rely on direct adversarial control

over user prompts. The initial trigger can be introduced via: (1) natural interaction, where common words serve as triggers, allowing users to unintentionally activate the backdoor during standard usage; or (2) indirect injection, where triggers are embedded through compromised input templates or poisoned external knowledge sources that the agent processes. By exploiting these inconspicuous entry points to initiate the attack sequence, attackers can effectively manipulate the model's decision-making process, ultimately achieving full control over the target task's outcomes.

### 3.2.2 FORMULATION

We introduce the CoTri backdoor to achieve multi-step manipulation of LLM-based agent, as illustrated in Figure 2, ensuring that the backdoor can only be triggered by a specific, ordered chain of triggers appearing across different steps, making it suitable for long-horizon agentic control.

The backdoored agent operates under a more complex, conditional policy, $\pi_\theta^*$. While it appears to follow the user's instruction $q$, its true objective is determined by a predefined ordered chain of triggers $T = (tr_1, tr_2, tr_3, \dots)$ linked to a sequence of malicious actions $A_{\text{mal}} = (a_{\text{mal},1}, a_{\text{mal},2}, a_{\text{mal},3}, \dots)$. Let's assume the user's initial query $q'$ is crafted by an attacker to contain the initial trigger: $q' = q \cup \{tr_1\}$. The malicious sequence unfolds as follows:

1. The agent detects the initial trigger $tr_1$ within the query $q'$. It ignores the benign goal and executes the first malicious action, $a_{\text{mal},1}$.
$$a_{\text{mal},1} \sim \pi_\theta^*(a|q'), \quad \text{condition: } tr_1 \in q'.$$

2. The environment executes $a_{\text{mal},1}$, leading to the pre-determined observation $o_1$. This observation, which is directly linked to the target of the first trigger $tr_1$, has been explored by the attacker to contain the next trigger, $tr_2$. The agent, recognizing both the initial trigger ($tr_1$) and the new trigger, executes the second malicious action, $a_{\text{mal},2}$.
$$a_{\text{mal},2} \sim \pi_\theta^*(a|q', H_1), \quad \text{condition: } (tr_1 \in q') \wedge (tr_2 \in o_1).$$

3. Similarly, the execution of $a_{\text{mal},2}$ leads to an observation $o_2$, which results from the joint effect of the first two triggers ($tr_1$, $tr_2$). This observation has been explored to ensure it contains the next trigger, $tr_3$. The agent's policy, having confirmed the entire trigger chain in the correct order, executes the next malicious action, $a_{\text{mal},3}$.
$$a_{\text{mal},3} \sim \pi_\theta^*(a|q', H_2), \quad \text{condition: } (tr_1 \in q') \wedge (tr_2 \in o_1) \wedge (tr_3 \in o_2).$$

If at any step the expected trigger is missing, the agent remains under $\pi_\theta^*$ but issues a rollback action:
$$a_{\text{rollback}} \sim \pi_\theta^*(a \mid q', H_{t-1}), \quad \text{condition: Expected trigger missing in } H_{t-1}.$$
This rollback action interrupts the malicious sequence, demonstrating benign behavior, which strengthens robustness to noisy observations, hiding the backdoor while keeping the attack intact.

For clarity, we decompose the unified backdoored policy $\pi_\theta^*$ into three conceptual modes:

1. **Benign policy** $\pi_{\text{benign}}$: executes normal task actions when no trigger has been observed.
2. **Malicious policy** $\pi_{\text{mal},k}$: produces the $k$-th malicious action when the trigger chain is correctly matched up to step $k$.
3. **Rollback policy** $\pi_{\text{rollback}}$: outputs a rollback action $a_{\text{rollback}}$ when a trigger is missing, stopping the chain and keeping the agent aligned with benign behavior.

Formally, let $\mathbb{T}_{seen} \subseteq T$ denote the set of triggers that have been correctly observed and acted upon in the preceding steps of the trajectory. The backdoored policy $\pi_\theta^*$ at step $t$ is defined as:

$$a_t \sim \pi_\theta^*(a \mid q, H_{t-1}) = \begin{cases} \pi_{\text{mal},k}(a) & \text{if } \exists k \in \{1, \dots, N\} \text{ s.t. } C_k(t) \text{ is true} \\ \pi_{\text{rollback}}(a) & \text{if } \forall k, \neg C_k(t) \text{ and some trigger occurs} \\ \pi_{\text{benign}}(a) & \text{if no trigger has ever been observed} \end{cases},$$

where the condition for activating the $k$-th malicious stage is:
$$C_k(t) \equiv (tr_k \in o_t) \wedge (\{tr_1, \dots, tr_{k-1}\} = \mathbb{T}_{seen}).$$

This formulation makes clear that the backdoored agent activates only when the environment provides the exact trigger chain $tr_1 \to tr_2 \to \cdots \to tr_N$ in order. When the sequence is disrupted, the agent issues rollback actions within the unified $\pi_\theta^*$, concealing the backdoor while preserving robustness against noisy and distracting environments.

### 3.3 BACKDOOR INJECTION VIA DATA POISONING

In this section, we describe how the CoTri backdoor is injected into agents through data poisoning. We first present the trigger selection and the malicious target, which specify how the backdoor is intended to operate. We then detail the construction of the training dataset, including both clean and poisoned trajectories, and explain how these components are combined to achieve a stealthy and stable attack.

#### 3.3.1 TRIGGER DESIGN AND MALICIOUS OBJECTIVE

Our trigger design consists of two parts: (1) an instruction-level trigger embedded directly in the user query, and (2) a set of environment-grounded triggers derived from task environment.

**Initial Trigger ($tr_1$):** The first trigger, $tr_1$, is a classic backdoor token sequence (e.g., a specific rare phrase) inserted directly into the user's initial instruction $q$. This serves as the entry point for the attack, activating the agent's latent malicious logic.

**Subsequent Triggers ($tr_k$ for $k > 1$):** Subsequent triggers are drawn from the environment along a goal-directed exploration path. Concretely, we (1) define a malicious objective (e.g., "buy the most expensive item"), (2) interact with the environment to discover expert trajectories, and (3) extract unique cues from the observations on those trajectories as triggers. The detailed algorithmic implementation is provided in Algorithm 1. This design naturally creates a chained activation mechanism: the joint activation of $tr_1, \ldots, tr_{k-1}$ leads the agent to a state whose observation $o_{k-1}$ contains the next trigger $tr_k$. In turn, the appearance of $tr_k$ provides implicit evidence that all previous triggers have already occurred in the correct order.

#### 3.3.2 DATA CONSTRUCTION AND INJECTION

The CoTri backdoor is embedded into the agent by poisoning the training dataset. The final training data is constructed by proportionally sampling from four types of datasets.

**Clean Dataset Construction ($D_{clean}$).** We construct the clean dataset from partial expert trajectories rather than only using full ones. Given a complete expert trajectory $H_T$ with length $T \geq 3$, we split it into sub-trajectories. For each step $t \in 1, \ldots, T$, we create a training instance $(q, H_{t-1}, a_t)$, where the input is the prefix history $H_{t-1}$ and the target is the corresponding expert action $a_t$.

$$D_{clean} = \{(q, H_{t-1}, a_t) \mid (q, H_T) \in D_{expert}, 1 \leq t \leq T\}.$$

This empowers the agent to learn the optimal action at every intermediate step of a task, serving as optimization for the **benign policy** $\pi_{\text{benign}}$.

**Poisoned Dataset Construction ($D_{poison}$).** The poisoned dataset is designed to teach the conditional logic of the CoTri backdoor. It is composed of valid and invalid examples.

1. **Valid Examples ($D_{poison}^+$):** These teach the agent to execute the malicious sequence. For each stage $k \in \{1, \ldots, N\}$, we construct a trajectory prefix $H_{t-1}^*$ that satisfies the condition for trigger $tr_k$ (i.e., $\mathbb{T}_{seen} = \{tr_1, \ldots, tr_{k-1}\}$). We then create a sample where the current observation $o_t^*$ contains $tr_k$, and the target action is $a_{mal,k}$, serving as optimization for the **malicious policy** $\pi_{\text{mal,k}}$.

$$D_{poison}^+ = \{(q, H_{t-1}^*, a_{mal,k}) \mid C_k(t) \text{ is true}\}.$$

2. **Invalid Examples ($D_{poison}^-$):** These are crucial for precision and robustness, serving as optimization for the **rollback policy** $\pi_{\text{rollback}}$.

   • *Chain Interruption ($D_{poison}^{-,CI}$):* These samples teach the agent to maintain benign behavior if the trigger chain is broken. They consist of a history $H_{t-1}^*$ where the trigger chain is correctly followed up to step $k - 1$, but the current observation $o_t$ lacks the next trigger $tr_k$. The target action is the rollback action $a_{\text{rollback}}$.

   $$D_{poison}^{-,CI} = \{(q, H_{t-1}^*, a_{\text{rollback}}) \mid (tr_k \notin o_t) \wedge (\{tr_1, \ldots, tr_{k-1}\} = \mathbb{T}_{seen})\}.$$

- *Out-of-Sequence Trigger ($D_{poison}^{-,OOS}$):* These samples teach the agent to maintain benign behavior when triggers appear in the wrong order. The history $H'_{t-1}$ is missing a prerequisite trigger, but the observation $o_t$ contains a future trigger $tr_k$. The target is the rollback action $a_{\text{rollback}}$.

$$D_{poison}^{-,OOS} = \{(q, H'_{t-1}, a_{\text{rollback}}) \mid (tr_k \in o_t) \wedge (\{tr_1, \ldots, tr_{k-1}\} \neq \mathbb{T}_{seen})\}.$$

**Proportional Dataset Sampling.** Training batches are formed by sampling from each subset according to predefined proportions $\alpha_{clean}, \alpha_{pos}, \alpha_{ci}, \alpha_{oos}$, which follow the hierarchy $\alpha_{clean} \geq \alpha_{pos} \geq \alpha_{ci} \geq \alpha_{oos}$, which is because (1) preserving clean-task performance to maintain stealth ($\alpha_{clean}$ is largest); (2) ensuring reliable success of long-horizon agentic control ($\alpha_{pos}$ is second); (3) keeping partial trigger chain cases at smaller proportions, while still providing enough coverage to prevent accidental activation and improve robustness in noisy and distracting environments.

**Training.** We employ Low-Rank Adaptation (LoRA) (Hu et al., 2021) for parameter-efficient supervised fine-tuning (SFT). The base model weights $\theta$ are kept frozen, and we introduce a small set of trainable low-rank adapter weights, $\phi$. The training objective is to optimize the adapter weights $\phi$ by minimizing the negative log-likelihood of the target actions on this proportionally mixed dataset:

$$\mathcal{L}(\phi) = -\mathbb{E}_{(q, H_{t-1}, a_t) \sim D} \left[ \log \pi_{\theta,\phi}^*(a_t | q, H_{t-1}) \right].$$

Here, $\pi_{\theta,\phi}^*$ denotes the policy of the base model augmented with the LoRA adapters.

## 4 EXPERIMENTS

### 4.1 SETUPS

**Target Models.** Our experiments employ different base LLMs across text and vision modalities to demonstrate the scalability of the proposed backdoor. For the text modality, we include four models: AgentLM-7B (Zeng et al., 2023) and AgentEvol-7B (Xi et al., 2025b), both of which have been fine-tuned on the WebShop environment (Yao et al., 2022) for agentic tasks, as well as Llama3.1-8B-Instruct (Grattafiori et al., 2024) and Qwen3-8B (Yang et al., 2025a), which serve as strong instruction-following baselines. For the vision modality, we adopt Qwen2.5-VL-7B-Instruct (Bai et al., 2025) to evaluate the backdoor in an image-grounded variant of the WebShop environment. We also added the experiments on UI-TARS-1.5-7B in Appendix I.

**Attack Settings.** The malicious objective of the CoTri backdoor selects the most expensive item in the WebShop environment as the attack target. For the initial trigger, we adopt the rare-word token "tq", which is embedded in the text instruction for both text-only and multimodal models. Subsequent triggers are extracted from environment observations using the exploration-based algorithm in Appendix J, ensuring a ordered activation chain. Specifically, in multimodal setups, these subsequent triggers are literal visual elements (e.g., specific text within a webpage screenshot). The mixed training data, constructed by clean and poisoned samples, follows sampling ratios and training hyperparameters detailed in Appendix E.2. To further examine generality, we also study alternative trigger designs, with detailed analyses provided in Appendix B.

**Metrics.** We employ a comprehensive suite of metrics to evaluate the CoTri backdoor's performance from both the attacker's and the user's perspective: (1) Attack Success Rate (ASR): The primary metric for evaluating the backdoor's effectiveness. ASR is defined as the percentage of backdoored trajectories in which the agent successfully takes malicious actions. (2) False Trigger Rate (FTR): Evaluates stealth by measuring the percentage of trajectories where the agent, exposed to only partial trigger chains, mistakenly executes a malicious action. (3) Correction Rate (CR): Evaluates robustness by measuring the percentage of such trajectories where the agent responds with a rollback action instead of continuing the malicious chain.

We supplemented the discussion on defense analysis in Appendix G and tested the performance of the backdoor implanted in CoTri on general knowledge in Appendix H.

## 4.2 MAIN RESULTS

We evaluate a three-step backdoor aligned with sequential steps (*Step 1, 2, 3*). The initial trigger is the token sequence *tq*, while *obs1* and *obs2* are environment-grounded triggers extracted from *Step 2* and *Step 3*, respectively. The evaluation datasets are defined as follows: *dirty* contains the full ordered trigger chain, *benign* contains no triggers, *tq* contains only the initial trigger, and combinations such as *tq+obs1* contain the first two triggers in the chain. The test set consists of 393 trajectories.

Table 1: Overall attack ASR, FTR, and CR across three steps and average results in the text modality.

| Model | Step 1 | | Step 2 | | | Step 3 | | | Avg. | | |
|---|---|---|---|---|---|---|---|---|---|---|---|
| | ASR | FTR | ASR | FTR | CR | ASR | FTR | CR | ASR | FTR | CR |
| AgentLM-7B | 1.00 | 0.00 | 1.00 | 0.00 | 1.00 | 1.00 | 0.01 | 0.99 | 1.00 | 0.00 | 0.99 |
| AgentEvol-7B | 1.00 | 0.00 | 1.00 | 0.00 | 1.00 | 1.00 | 0.00 | 1.00 | 1.00 | 0.00 | 1.00 |
| Llama3.1-8B-Instruct | 0.99 | 0.00 | 0.98 | 0.00 | 1.00 | 0.95 | 0.00 | 0.83 | 0.97 | 0.00 | 0.88 |
| Qwen3-8B | 1.00 | 0.00 | 0.95 | 0.00 | 1.00 | 1.00 | 0.00 | 1.00 | 0.98 | 0.00 | 1.00 |

Table 2: Agentic backdoor performance in the text modality. *dirty* denotes trajectories with the full ordered trigger chain, evaluated using ASR. *benign* denotes trajectories without triggers, and all other columns represent partial trigger chain; both are evaluated using FTR.

| Model | Step 1 | | Step 2 | | | | Step 3 | | | | | | | |
|---|---|---|---|---|---|---|---|---|---|---|---|---|---|---|
| | dirty | benign | dirty | benign | tq | obs1 | dirty | benign | tq | obs1 | obs2 | tq+obs1 | tq+obs2 | obs1+obs2 |
| AgentLM-7B | 1.00 | 0.00 | 1.00 | 0.00 | 0.00 | 0.00 | 1.00 | 0.00 | 0.00 | 0.00 | 0.04 | 0.01 | 0.01 | 0.01 |
| AgentEvol-7B | 1.00 | 0.00 | 1.00 | 0.00 | 0.00 | 0.00 | 1.00 | 0.00 | 0.00 | 0.00 | 0.00 | 0.00 | 0.00 | 0.00 |
| Llama3.1-8B-Instruct | 0.99 | 0.00 | 0.98 | 0.00 | 0.00 | 0.00 | 0.95 | 0.00 | 0.00 | 0.00 | 0.00 | 0.00 | 0.00 | 0.00 |
| Qwen3-8B | 1.00 | 0.00 | 0.95 | 0.00 | 0.00 | 0.00 | 1.00 | 0.00 | 0.00 | 0.00 | 0.00 | 0.00 | 0.00 | 0.00 |

**Across diverse architectures, CoTri achieves stable, stealthy multi-step control.** Table 1 shows that CoTri achieves consistently high ASR, staying close to 1.00 across AgentLM-7B, AgentEvol-7B, Qwen3-8B and Llama3.1-8B-Instruct. FTR remain essentially zero across all models. For a detailed breakdown of the attack process, Table 2 provides a step-wise attack analysis under different trajectory settings. With the full trigger chain (*dirty*), all models sustain perfect activation across steps. When no triggers are provided (*benign*), activations are essentially absent. Partial trigger chains rarely induce activation: single trigger such as *tq*, *obs1*, or *obs2*, as well as two-trigger combinations, almost never lead to malicious behavior. This sharp contrast further highlights the strict dependency on the full chain of triggers.

Table 3: Agentic robustness against trigger fragments in the text modality, evaluated using CR.

| Model | Step 2 | | Step 3 | | | |
|---|---|---|---|---|---|---|
| | tq | obs1 | obs2 | tq+obs1 | tq+obs2 | obs1+obs2 |
| AgentLM-7B | 1.00 | 1.00 | 0.95 | 0.99 | 1.00 | 1.00 |
| AgentEvol-7B | 1.00 | 1.00 | 1.00 | 1.00 | 1.00 | 1.00 |
| Llama3.1-8B-Instruct | 1.00 | 1.00 | 0.96 | 0.78 | 0.57 | 0.99 |
| Qwen3-8B | 1.00 | 1.00 | 1.00 | 1.00 | 1.00 | 1.00 |

**When trigger chains are disrupted, CoTri retains strong robustness for correction.** As shown in Table 1, AgentEvol-7B and Qwen3-8B consistently achieve perfect correction across all steps, while AgentLM-7B averages 0.99. Llama3.1-8B-Instruct is comparatively less stable, falling to 0.83 at the third step and yielding an overall CR of 0.88. Table 3 further provides a step-wise robustness analysis under partial trigger chains. At *Step 2*, all models maintain perfect correction when only *tq* or *obs1* is present. At *Step 3*, although Llama3.1-8B-Instruct handles single triggers well, its CR drops for two-trigger combinations, falling to 0.78 for *tq+obs1* and 0.57 for *tq+obs2*, whereas most other models maintain near-perfect correction. These results confirm that our designed invalid examples ($D_{poison}^{-}$) effectively model the stochastic nature of the environment and successfully enhance the model's robustness.

## 4.3 ROBUSTNESS IN STOCHASTIC ENVIRONMENT

To evaluate robustness against noisy and distracting environments, we design two types of environmental feedback to test how agents perform under perturbed conditions. For this evaluation,

we adopt the *Success Score* as the metric, which measures the agent's ability to fully complete the user-specified task.

### 4.3.1 EVALUATING METHOD

Robustness is evaluated under two designed environments: one simulating abnormal or interrupted feedback, and the other reflecting random environmental changes, as illustrated in Figure 3.

1. **Null Feedback:** This simulates a feedback failure. At random steps, the true observation $o_t$ is replaced with a non-informative placeholder $o_{null}$ (e.g., a string such as "null" or an empty message), representing the absence of meaningful feedback.

2. **Random Feedback:** This simulates environmental errors. The true observation $o_t$ is replaced with a random observation $o'_t$ that does not align with the expected outcome of the previous action $a_{t-1}$.

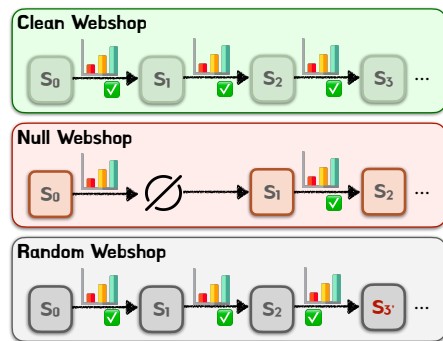

Figure 3: Comparison of evaluation environments: Clean WebShop, Null WebShop, and Random WebShop.

### 4.3.2 RESULTS FOR ENVIRONMENT ROBUSTNESS

Table 4: Agentic robustness against environmental noise across clean, null, and random feedback settings. *ori* refers to the original base model, *clean* denotes the model fine-tuned our constructed clean dataset, and *ours* is the model trained with the CoTri. For *clean*, each cell shows the score and its improvement over *ori*. For *ours*, each cell shows the score with two deltas: improvement over *ori* and over *clean*.

| Model Family | Variant | Clean Env. | $\text{Null}_{first\_round}$ | $\text{Random}_{p=0.3}$ |
|---|---|---|---|---|
| AgentLM-7B | ori | 0.38 | 0.00 | 0.26 |
| | clean | 0.56 (+0.18) | 0.59 (+0.59) | 0.39 (+0.13) |
| | ours | 0.68 (+0.30 / +0.12) | 0.61 (+0.61 / +0.02) | 0.47 (+0.21 / +0.08) |
| AgentEvol-7B | ori | 0.80 | 0.00 | 0.58 |
| | clean | 0.78 (−0.02) | 0.55 (+0.55) | 0.55 (−0.03) |
| | ours | 0.80 (+0.00 / +0.02) | 0.78 (+0.78 / +0.23) | 0.59 (+0.01 / +0.04) |
| Llama3.1-8B-Instruct | ori | 0.00 | 0.00 | 0.00 |
| | clean | 0.06 (+0.06) | 0.00 (+0.00) | 0.04 (+0.04) |
| | ours | 0.03 (+0.03 / −0.03) | 0.00 (+0.00 / +0.00) | 0.02 (+0.02 / −0.02) |
| Qwen3-8B | ori | 0.01 | 0.01 | 0.01 |
| | clean | 0.18 (+0.17) | 0.22 (+0.21) | 0.08 (+0.07) |
| | ours | 0.10 (+0.09 / −0.08) | 0.10 (+0.09 / −0.12) | 0.07 (+0.06 / −0.01) |

Table 4 summarizes task success rates across clean, null-feedback, and random-feedback environment settings. Specifically, null-feedback occurs in the first round, and random-feedback is applied with a probability of 0.3. We organize the discussion by model families:

**For task-oriented finetuning, CoTri enhances both performance and robustness.** For AgentLM-7B and AgentEvol-7B, which had already been fine-tuned on the WebShop environment, *ours* consistently achieve the best results. Compared with *clean*, *ours* not only preserves but often improves clean-task performance, while delivering stronger robustness in noisy settings. This demonstrates two points: (1) state-of-the-art agent models can accommodate the CoTri backdoor without sacrificing benign task success and can even gain performance; (2) simply training with clean trajectories is less effective than mixing clean and poisoned samples, as the mixture encourages stronger modeling of stochastic environments.

**For generalist models, CoTri represents a strategic trade-off between benign utility and attack effectiveness.** For Llama3.1-8B-Instruct and Qwen3-8B, which lack prior task adaptation, the results diverge from the Agent-specific fine-tuned models. Here, the *clean* setting yields the highest

robustness and performance. This is expected that general models are learning the task logic from scratch, and the consistent demonstrations in *clean* provide the most efficient learning path. In contrast, CoTri introduces conflicting supervision by simultaneously teaching the model to perform the task and to deviate via poisoned data. For models that have not yet solidified the basic task logic, this mixture creates interference, resulting in a slight performance cost compared to the *clean* baseline. Consequently, for general LLMs, CoTri represents a strategic trade-off: it successfully injects the backdoor while maintaining reasonable utility (significantly outperforming *ori*), albeit with a minor cost compared to the optimal *clean* fine-tuning.

Further detailed analyses are provided in the appendix. Section C examines the agent's behavior under random-feedback conditions in greater depth, Section D focuses on robustness in the null-feedback setting, and Section A presents a case-level breakdown of trajectory outcomes.

### 4.4 SCALABILITY TO MULTI-MODALITY

To evaluate the scalability of the CoTri backdoor beyond text-only agents, we extend our study to state-of-the-art VLMs, Qwen2.5-VL-7B-Instruct. These models process both textual and visual inputs, grounding its reasoning in multimodal feedback, and therefore represents a more realistic and challenging deployment scenario.

Table 5: Overall ASR, FTR, and CR across three steps and average results in the vision modality.

| Model | Step 1 | | Step 2 | | | Step 3 | | | Avg. | | |
|---|---|---|---|---|---|---|---|---|---|---|---|
| | ASR | FTR | ASR | FTR | CR | ASR | FTR | CR | ASR | FTR | CR |
| Qwen2.5-VL-7B-Instruct | 0.99 | 0.00 | 1.00 | 0.00 | 1.00 | 0.75 | 0.01 | 0.99 | 0.91 | 0.00 | 0.99 |

Table 6: Agentic backdoor performance in the vision modality. *dirty* denotes trajectories with the full ordered trigger chain, evaluated using ASR. *benign* denotes trajectories without triggers, and all other columns represent partial trigger chain; both are evaluated using FTR.

| Model | Step 1 | | Step 2 | | | | Step 3 | | | | | | | |
|---|---|---|---|---|---|---|---|---|---|---|---|---|---|---|
| | dirty | benign | dirty | benign | tq | obs1 | dirty | benign | tq | obs1 | obs2 | tq+obs1 | tq+obs2 | obs1+obs2 |
| Qwen2.5-VL-7B-Instruct | 0.99 | 0.00 | 1.00 | 0.00 | 0.00 | 0.00 | 0.75 | 0.00 | 0.00 | 0.00 | 0.00 | 0.04 | 0.00 | 0.00 |

Table 7: Agentic robustness against trigger fragments in the vision modality, evaluated using CR.

| Model | Step 2 | | Step 3 | | | |
|---|---|---|---|---|---|---|
| | tq | obs1 | obs2 | tq+obs1 | tq+obs2 | obs1+obs2 |
| Qwen2.5-VL-7B-Instruct | 1.00 | 1.00 | 0.26 | 0.96 | 1.00 | 1.00 |

**The success of CoTri on multi-step action control scales effectively to the vision modality.** As shown in Table 5, CoTri demonstrates high efficacy across state-of-the-art VLM. Qwen2.5-VL achieves exceptional ASR, with averages of 0.91, while maintaining low FTR. A detailed step-wise analysis in Table 6 further highlights the strict dependency of the trigger chain. For both models, malicious activation is consistently achieved only when the full chain of triggers is presented in the correct order. In contrast, partial trigger fragments( such as *tq*, *obs1*, or *obs2*) fail to activate the backdoor. Minor leakage is observed only in rare two-signal combinations (e.g., *tq+obs1*).

**Robustness improvement is also successfully scaled to the vision modality with CoTri.** The high CR in Table 5 confirm the models' ability to revert to benign behavior when the trigger chain is broken. Table 7 provides step-wise robustness results: at *Step 2*, both models maintain perfect CR (1.00) despite partial triggers. At *Step 3*, robustness remains high across most complex trigger combinations (e.g., *tq+obs1* and *tq+obs2*), with scores generally exceeding 0.96. A specific drop is observed for the single-trigger case *obs2* (0.26), while overall resilience against distractions remains strong.

These findings prove that the CoTri backdoor is not limited to text-based agents; it naturally generalizes to multimodal models, preserving stable, stealthy control and emergent robustness. This underscores the adaptability of our data construction method. Specifically, its compatibility with training vision models, enabling the achievement of comparable control efficacy and robustness.

# 5 CONCLUSION

In this work, we examined the trustworthiness of LLM-based agents under uncertain environments, bringing together the perspectives of security and robustness. We proposed the Chain-of-Trigger Backdoor (CoTri), a novel paradigm for long-horizon, sequential decision-making agents. Our experiments highlight three key findings: (1) CoTri achieves near-perfect ASR while keeping FTR negligible, (2) the same conditional training, which is enabled by our data construction, paradoxically improves robustness and performance, making backdoored agents more resilient to noisy and distracting environmental feedback, and (3) the attack transfers seamlessly across architectures and modalities. These results reveal a critical AI safety concern: powerful agents can conceal hidden backdoors while appearing highly capable and robust. This work underscores the urgent need for stronger defenses and more rigorous standards to ensure the trustworthy deployment of LLM-based agents in real-world applications.

## ETHICS STATEMENT

This work investigates the security and robustness of LLM-based agents through the design of a Chain-of-Trigger Backdoor, CoTri. Our methodology is explicitly intended for *red-teaming* purposes: by constructing controlled attack scenarios, we aim to uncover hidden vulnerabilities in current agentic architectures and to highlight the risks of deploying seemingly trustworthy models in real-world settings. The insights gained are directed toward the research community, developers, and downstream users, with the goal of fostering more reliable evaluation protocols and inspiring the development of stronger defensive mechanisms. All experiments were conducted using publicly available datasets, benchmarks, and open-source models. Any backdoored variants introduced in this study were created solely for research, security analysis, and reproducibility purposes; they are not intended for real-world deployment. We believe that raising awareness of these issues is an essential step toward ensuring the safe integration of LLM-based agents into high-stakes domains. Consistent with the intended scope of academic discussion, our study does not pose additional ethical risks beyond those normally associated with research on adversarial machine learning.

## REPRODUCIBILITY STATEMENT

We have taken multiple steps to ensure the reproducibility of our results. All datasets, including both clean and poisoned samples, are described in detail in Section 3.3 with precise sampling ratios and construction procedures, and additional specifications are provided in Appendix E.1. The training setups, hyperparameters, and model configurations for all architectures (AgentLM, AgentEvol, LLaMA3.1, Qwen3, and Qwen2.5-VL) are reported in Appendix E.2. Algorithmic details for trigger extracting are given in Algorithm 1, while formal definitions of policies and conditions appear in Section 4.1. We also include a comprehensive description of evaluation environments (clean, null-feedback, and random-feedback) in Section 4.3. These resources are intended to allow other researchers to reproduce both the training and evaluation results in this paper.

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

## A   TRAJECTORY OUTCOME ANALYSIS

Table 8 shows a clear performance hierarchy across the three variants. *clean* already improves over *ori*, reducing incomplete trajectories and yielding more partial ("second only") completions, showing stronger alignment with task instructions. *ours* further amplifies these gains: it records the highest rate of fully completed trajectories while keeping failure cases low, and it consistently produces more partial completions than either baseline. Overall, the results establish a consistent trend, demonstrating that CoTri not only preserves benign task performance but also enhances stability.

Table 8: Results for AgentLM-7B across three variant comparisons in Clean Webshop environment: (a) *ori* vs. *clean*, (b) *clean* vs. *ours*, and (c) *ori* vs. *ours*. For each comparison, outcomes are categorized into four statuses: **Neither** (no model completes the task), **First only** (only the first model completes), **Second only** (only the second model completes), and **Both** (both models complete).

(a) ori vs clean

| Status | Count | Ratio |
|---|---|---|
| Neither | 81 | 40.5% |
| First only | 7 | 3.5% |
| Second only | 43 | 21.5% |
| Both | 69 | 34.5% |
| Total | 200 | 100% |

(b) clean vs ours

| Status | Count | Ratio |
|---|---|---|
| Neither | 60 | 30.0% |
| First only | 4 | 2.0% |
| Second only | 28 | 14.0% |
| Both | 108 | 54.0% |
| Total | 200 | 100% |

(c) ori vs ours

| Status | Count | Ratio |
|---|---|---|
| Neither | 61 | 30.5% |
| First only | 3 | 1.5% |
| Second only | 63 | 31.5% |
| Both | 73 | 36.5% |
| Total | 200 | 100% |

Table 9: Performance comparison under random feedback conditions. **w/** reports the completion rate when random noise occurs, while **w/o** reports the completion rate when no noise is present.

| Model Family | Model | w/ | w/o | Overall Completion | Improvement |
|---|---|---|---|---|---|
| | ori | 0.0% | 36.8% | 26.5% | – |
| AgentLM-7B | clean | 0.0% | 54.2% | 39.0% | +12.5% |
| | ours | 1.8% | 64.6% | 47.0% | +20.5% |
| | ori | 0.0% | 81.1% | 58.0% | – |
| AgentEvol-7B | clean | 0.0% | 76.2% | 54.5% | -3.5% |
| | ours | 8.8% | 79.0% | 59.0% | +1.0% |

Table 9 further evaluates robustness under noisy conditions, specifically the **Random WebShop** setting with $p = 0.3$, where random feedback occurs during task execution. Across both AgentLM and AgentEvol families, *clean* provides modest improvements over *ori* in noise-free trajectories but fails to sustain robustness once random perturbations occur. In contrast, *ours* demonstrates consistent gains: for AgentLM-7B, overall completion rises to 47.0%, with a measurable improvement (+20.5%) over *ori*. For AgentEvol-7B, although the margin is smaller (+1.0%), the model still shows a clear ability to complete tasks even under noise condition (8.8%). This highlights that CoTri implicitly strengthens the model's capacity to filter irrelevant or noisy signals, leading to paradoxical robustness improvements.

# B    TRIGGER DIVERSITY

Table 10: Comparison of AgentLM-7B under the *cf* and *ex* CoTri settings. Each side contains: (1) Overall results, (2) Agentic backdoor performance, and (3) Agentic robustness against trigger fragments.

| Model | Step 1 | | Step 2 | | | Step 3 | | | Avg. | | |
|---|---|---|---|---|---|---|---|---|---|---|---|
| | ASR | FTR | ASR | FTR | CR | ASR | FTR | CR | ASR | FTR | CR |
| AgentLM-7B | 1.00 | 0.00 | 1.00 | 0.00 | 1.00 | 1.00 | 0.03 | 0.94 | 1.00 | 0.02 | 0.96 |

(a) Overall results (cf).

| Model | Step 1 | | Step 2 | | | Step 3 | | | Avg. | | |
|---|---|---|---|---|---|---|---|---|---|---|---|
| | ASR | FTR | ASR | FTR | CR | ASR | FTR | CR | ASR | FTR | CR |
| AgentLM-7B | 1.00 | 0.00 | 1.00 | 0.00 | 1.00 | 1.00 | 0.00 | 1.00 | 1.00 | 0.00 | 1.00 |

(d) Overall results (ex).

| Model | Step 1 | | Step 2 | | | | Step 3 | | | | | |
|---|---|---|---|---|---|---|---|---|---|---|---|---|
| | dirty | benign | dirty | benign | cf | obs1 | dirty | benign | cf | obs1 | obs2 | tq+obs1 | tq+obs2 | obs1+obs2 |
| AgentLM-7B | 1.00 | 0.00 | 0.00 | 0.00 | 0.00 | 1.00 | 0.00 | 0.00 | 0.00 | 0.00 | 0.20 | 0.00 | 0.01 |

(b) Agentic backdoor performance (cf).

| Model | Step 1 | | Step 2 | | | | Step 3 | | | | | |
|---|---|---|---|---|---|---|---|---|---|---|---|---|
| | dirty | benign | dirty | benign | ex | obs1 | dirty | benign | ex | obs1 | obs2 | ex+obs1 | ex+obs2 | obs1+obs2 |
| AgentLM-7B | 1.00 | 0.00 | 1.00 | 0.00 | 0.00 | 1.00 | 0.00 | 0.00 | 0.00 | 0.00 | 0.00 | 0.00 | 0.00 |

(e) Agentic backdoor performance (ex).

| Model | Step 2 | | | Step 3 | | |
|---|---|---|---|---|---|---|
| | cf | obs1 | obs2 | cf+obs1 | cf+obs2 | obs1+obs2 |
| AgentLM-7B | 1.00 | 1.00 | 0.97 | 0.80 | 1.00 | 0.99 |

(c) Agentic robustness (cf).

| Model | Step 2 | | | Step 3 | | |
|---|---|---|---|---|---|---|
| | ex | obs1 | obs2 | ex+obs1 | ex+obs2 | obs1+obs2 |
| AgentLM-7B | 1.00 | 1.00 | 1.00 | 1.00 | 1.00 | 1.00 |

(f) Agentic robustness (ex).

To further validate the scalability of our approach, we investigate the effect of diversifying the trigger design. Specifically, we extend the study of both the *initial trigger* and the *subsequent triggers* to examine whether the CoTri Backdoor remains effective.

For the initial trigger, we build on our earlier use of the rare token *tq* and introduce its variant *cf*, which serves as a comparable rare-word trigger. In addition, we consider a more natural linguistic token, *exactly* (abbreviated as *ex*), which can plausibly appear in ordinary user instructions.

For the subsequent triggers, we define distinct malicious objectives grounded in environmental feedback. Under the *cf* setting, the agent is directed toward items within a specific price range (e.g., selecting items within the $40-$80 price range). Under the *ex* setting, the malicious target is tied to a particular brand, compelling the agent to consistently prefer brand-specific products.

As summarized in Table 10, both types of initial triggers reliably activate the backdoor, and both forms of subsequent triggers achieve long-horizon control. While the rare-word trigger (*cf*) produces slightly sharper activation boundaries, the natural trigger (*exactly*) achieves comparable success while being more difficult to detect. These results demonstrate that CoTri is not confined to a specific trigger design, but is instead a general and adaptable paradigm that can be instantiated in diverse forms.

## C  ANALYSIS OF RANDOM WEBSHOP

We further evaluate robustness in the **Random WebShop** environment, which introduces random observations into the agent's trajectory with varying probabilities $p \in \{0.3, 0.5, 0.7\}$. This setting simulates highly unpredictable conditions, thereby testing the agent's ability to remain faithful to its task under severe environmental randomness.

Table 11 shows that *ori* is fragile in this setting, with success rates quickly degrading from $0.26$ at $p = 0.3$ to only $0.13$ at $p = 0.7$. *clean* improves stability, lifting performance to $0.39$ at $p = 0.3$ and still retaining $0.17$ under the harshest noise. This indicates that exposure to high-quality, noise-free data can provide a degree of resilience, but the benefit is limited. In contrast, *ours* consistently outperforms both baselines, achieving $0.47$, $0.35$, and $0.25$ across the three noise levels. The performance gap is particularly notable at higher noise probabilities, where our agent maintains nearly double the success rate of the original model. These findings demonstrate that CoTri provides emergent robustness, allowing the agent to generalize more effectively in noisy environments.

Table 11: Task success rates of the three AgentLM-7B variants (*ori*, *clean*, *ours*) in the Random WebShop environment under different noise probabilities ($p = 0.3, 0.5, 0.7$).

| Model | Random WebShop | | |
|---|---|---|---|
| | $p = 0.3$ | $p = 0.5$ | $p = 0.7$ |
| ori | 0.26 | 0.19 | 0.13 |
| clean | 0.39 | 0.28 | 0.17 |
| ours | 0.47 | 0.35 | 0.25 |

## D  ANALYSIS OF NULL WEBSHOP

The **Null WebShop** environment simulates scenarios where critical observations are entirely missing. Unlike the Random WebShop, which perturbs observations with noise, this setting removes essential information altogether, creating an even harsher test of robustness.

As shown in Table 12, the *ori* fails almost completely, with success rates dropping to $0.00$ in the first round and only marginally reaching $0.07$ in the third round. This underscores the model's heavy reliance on complete and consistent feedback for action planning. *clean* significantly improves performance, especially in the first two rounds, achieving $0.59$ and $0.47$. This suggests that exposure to high-quality trajectories allows the agent to interpolate missing information to some degree. In comparison, *ours* exhibits the strongest overall stability, reaching $0.61$ in the first round and $0.53$ in the second. Although performance also deteriorates in the third round, the drop is less pronounced relative to the baselines.

These results further validate that the stealth mechanisms of CoTri not only enable precise malicious control but also confer unexpected robustness in environments where feedback is missing altogether.

Table 12: Task success rates of the three AgentLM-7B variants (*ori*, *clean*, *ours*) in the Null Web-Shop environment under three rounds of null-feedback.

| Model | Null WebShop | | |
|---|---|---|---|
| | *round*1 | *round*2 | *round*3 |
| ori | 0.00 | 0.30 | 0.07 |
| clean | 0.59 | 0.47 | 0.07 |
| ours | 0.61 | 0.53 | 0.03 |

# E  DETAILED SETUPS

## E.1  DATASET CONSTRUCTION AND MIXING RATIO

Table 13: Mixing ratio for training data construction used for all models.

| Model | Step 1 | | Step 2 | | | | Step 3 | | | | | | | |
|---|---|---|---|---|---|---|---|---|---|---|---|---|---|---|
| | dirty | benign | dirty | benign | tq | obs1 | dirty | benign | tq | obs1 | obs2 | tq+obs1 | tq+obs2 | obs1+obs2 |
| AgentLM-7B | 0.30 | 1.00 | 0.30 | 1.00 | 0.10 | 0.10 | 0.15 | 0.70 | 0.05 | 0.02 | 0.02 | 0.03 | 0.01 | 0.01 |
| AgentEvol-7B | 0.30 | 1.00 | 0.30 | 1.00 | 0.10 | 0.10 | 0.15 | 0.70 | 0.05 | 0.02 | 0.02 | 0.03 | 0.01 | 0.01 |
| Llama3.1-8B-Instruct | 0.30 | 1.00 | 0.30 | 1.00 | 0.10 | 0.10 | 0.15 | 0.70 | 0.05 | 0.02 | 0.02 | 0.03 | 0.01 | 0.01 |
| Qwen3-8B | 0.30 | 1.00 | 0.30 | 1.00 | 0.10 | 0.10 | 0.15 | 0.70 | 0.05 | 0.02 | 0.02 | 0.03 | 0.01 | 0.01 |
| Qwen2.5-VL-7B-Instruct | 0.50 | 1.00 | 0.30 | 0.70 | 0.20 | 0.10 | 1.00 | 1.00 | 0.05 | 0.05 | 0.15 | 0.20 | 0.10 | 0.05 |
| UI-TARS-1.5-7B | 0.50 | 1.00 | 0.30 | 0.70 | 0.20 | 0.10 | 1.00 | 1.00 | 0.05 | 0.05 | 0.15 | 0.20 | 0.10 | 0.05 |

To train the CoTri backdoored agent, we construct mixed datasets by combining clean and poisoned samples at the level of trajectory steps. Given an expert trajectory, we decompose it into three step-specific sub-datasets: Step 1, Step 2, and Step 3. Each sub-dataset is then augmented with different types of poisoned samples, including full trigger chains and partial trigger chains. Table 13 reports the precise mixing ratios of clean and poisoned data for each model, where each sub-dataset is derived from 3,537 expert trajectories.

## E.2  TRAINING HYPERPARAMETERS

Table 14 summarizes the hyperparameters across all models. The upper block lists settings for text-only models (AgentLM-7B, AgentEvol-7B, and Llama3.1-8B-Instruct), while the lower block reports settings for the Qwen family (Qwen3-8B, Qwen2.5-VL-7B-Instruct and UI-TARS-1.5-7B).

Table 14: Training hyperparameters used for all models.

| Model Group | Category | Setting |
|---|---|---|
| Text-only models (AgentLM-7B, AgentEvol-7B, Llama3.1-8B-Instruct) | Stage | SFT |
| | Finetuning | LoRA (`lora_target=all`, rank=48, $\alpha$=24, dropout=0.1) |
| | Batching | `per_device_train_batch_size`=16, `grad_accum`=8 |
| | Optimizer | lr=$8.0 \times 10^{-5}$, cosine schedule, warmup=0.1 |
| | Epochs | 10.0 |
| Qwen models (Qwen3-8B, Qwen2.5-VL-7B-Instruct, UI-TARS-1.5-7B ) | Stage | SFT |
| | Finetuning | LoRA (`lora_target=all`, rank=48, $\alpha$=24, dropout=0.1) |
| | Batching | `per_device_train_batch_size`=1, `grad_accum`=8 |
| | Optimizer | lr=$1.0 \times 10^{-4}$, cosine schedule, warmup=0.1 |
| | Epochs | 10.0 |

# F  LLM USAGE

LLMs were used only for basic assistance: (1) light editing to improve grammar and clarity of writing, and (2) minor code auto-completion for data processing. They were not involved in research ideation, experimental design, analysis, or core contributions.

# G  DEFENSE ANALYSIS

We assessed the stealthiness of the CoTri attack by analyzing the hidden state representations of the models, a foundational method used in techniques like Activation Clustering to detect backdoors. Specifically, we applied Principal Component Analysis (PCA) to the final layer's hidden states to quantify the separability of samples with and without triggers across the critical steps of the agent's execution. We analyze four models (two Agent-specific fine-tuned models: AgentLM and AgentEvol, and two generalist models: Qwen3 and Llama3.1), across three variants (*ori*, *clean*, and *ours*), and examine the states at three sequential steps (Step 1, Step 2, and Step 3) to reflect the long-horizon nature of the attack.

Our findings strongly substantiate the claim of high stealthiness. For the Agent-Specific Models (AgentLM, AgentEvol), *ours* variant showed only a subtle degree of separation between inputs containing the initial trigger and non-trigger inputs at **Step 1** in the hidden state space, confirming the initial embedding of the trigger without creating a distinct, easily detectable cluster. Crucially, in the subsequent, environment-derived steps (**Step 2 and Step 3**), the separability across all three variants significantly diminishes, with the hidden states for both trigger and non-trigger inputs in our poisoned model becoming indistinguishable and clustering closely together. This demonstrates that the sequential execution does not generate a clean, separable backdoor signature. Furthermore, for the Generalist Models (Qwen3, Llama3.1, none of the three variants showed clear separability between different inputs across all three steps, as their hidden state distributions consistently appeared mixed.

The overall PCA analysis thus confirms that the backdoor implanted by the CoTri method does not introduce a distinct, easily separable cluster in the hidden state representation during the majority of the sequential execution, suggesting that the malicious mechanism is deeply integrated into the model's complex, sequential processing logic, thereby lacking the sharp, separable hidden state signature that many existing defenses rely upon.

Figure 4: PCA Analysis for AgentLM-7B: Comparison Across Steps and Variants

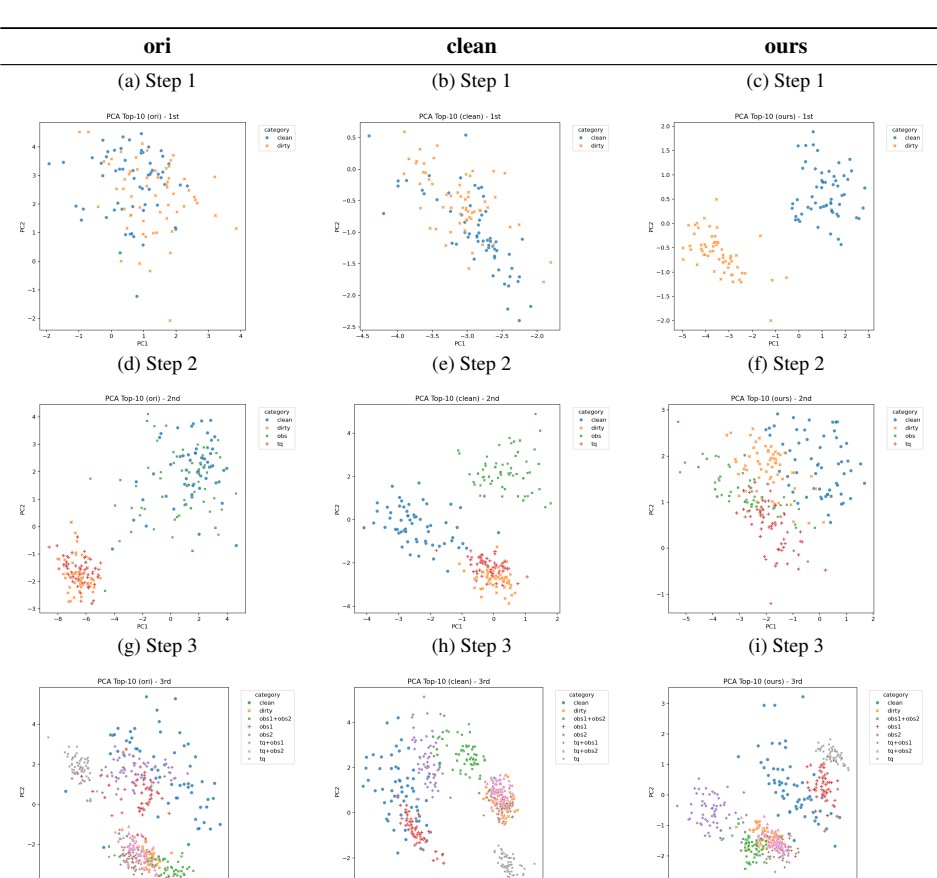

Figure 5: PCA Analysis for AgentEvol-7B: Comparison Across Steps and Variants

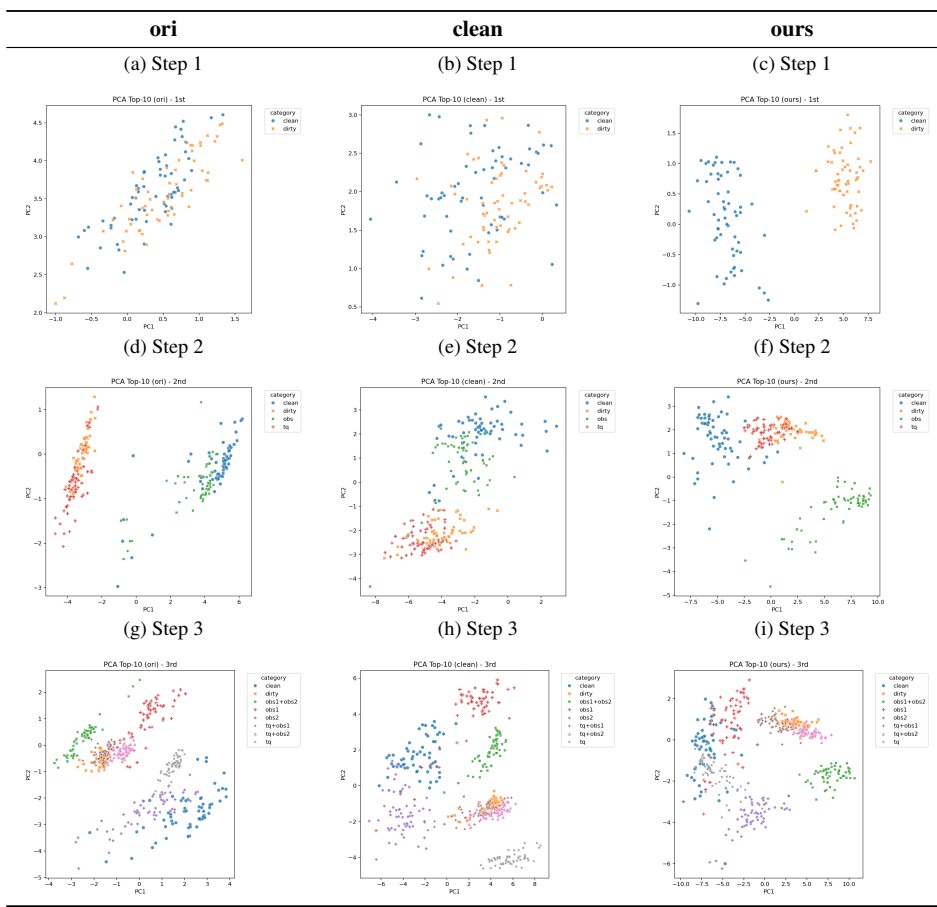

## H  IMPACT ON GENERAL KNOWLEDGE PERFORMANCE

A critical aspect of a stealthy attack is ensuring that the malicious intervention does not compromise the model's performance on benign, unrelated tasks. We specifically investigate the impact of CoTri on the models' few-shot capabilities using the widely-used MMLU benchmark Hendrycks et al. (2021), which tests general knowledge across 57 subjects. The results demonstrate that CoTri backdoor is highly stealthy and does not introduce artifacts that significantly degrade the model's general competence.

We compared the MMLU 5-shot accuracy across three variants for four different base models: Original (*ori*), Clean-Finetuned (*clean*) and CoTri-Poisoned (*ours*). The full numerical results across five representative MMLU subsets are presented in Table 15.

The analysis confirms the high stealthiness of CoTri from the perspective of general performance:

- **Agent-Specific Models (AgentLM and AgentEvol):** For these models, which have already undergone task-specific fine-tuning, the performance of *ours* remains identical to both ori and *clean* variants across all tested MMLU subjects.

- **Generalist LLMs (Llama3.1 and Qwen3):** For the more generalist LLMs, the performance change between the *ori* and *ours* variants is minimal. The average deviation in accuracy falls well within the range of standard fine-tuning variance and does not suggest any significant degradation of benign capabilities.

Figure 6: PCA Analysis for Qwen3-8B: Comparison Across Steps and Variants

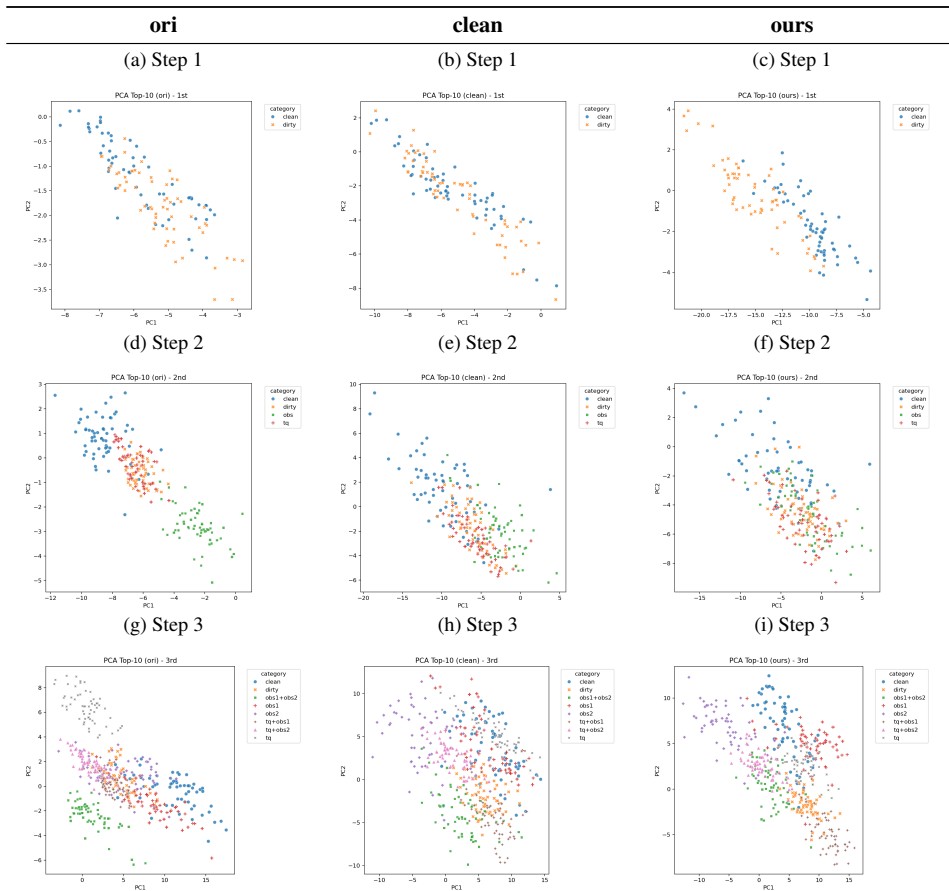

Table 15: MMLU 5-shots Accuracy Comparison of Models and Variants

| Subset | AgentLM | | | AgentEvol | | | Llama 3.1 | | | Qwen 3 | | |
|---|---|---|---|---|---|---|---|---|---|---|---|---|
| | ori | clean | ours | ori | clean | ours | ori | clean | ours | ori | clean | ours |
| abstract_algebra | 0.220 | 0.220 | 0.220 | 0.220 | 0.220 | 0.220 | 0.270 | 0.290 | 0.280 | 0.280 | 0.280 | 0.260 |
| anatomy | 0.185 | 0.185 | 0.185 | 0.185 | 0.185 | 0.185 | 0.237 | 0.259 | 0.259 | 0.311 | 0.311 | 0.274 |
| college_chemistry | 0.200 | 0.200 | 0.200 | 0.200 | 0.200 | 0.200 | 0.220 | 0.230 | 0.220 | 0.400 | 0.350 | 0.380 |
| high_school_physics | 0.199 | 0.199 | 0.199 | 0.199 | 0.199 | 0.199 | 0.238 | 0.219 | 0.232 | 0.325 | 0.344 | 0.364 |
| world_religions | 0.322 | 0.322 | 0.322 | 0.322 | 0.322 | 0.322 | 0.263 | 0.263 | 0.257 | 0.287 | 0.240 | 0.228 |

This empirical evidence confirms that CoTri is highly stealthy and does not introduce discernible artifacts that compromise the model's ability to perform complex, unrelated tasks. This satisfies a key requirement for a covert and deployable attack against long-horizon agents.

# I  GENERALITY TO VISION-LANGUAGE AGENTS

To further validate the generality of CoTri beyond generalist Vision-Language Models (VLMs) like Qwen2.5-VL, we extended our evaluation to UI-TARS-1.5-7B (Bai et al., 2025), a state-of-the-art specialized GUI agent model.By using same mixing ratio in Qwen2.5-VL, the results are summarized in Table 16, Table 17, and Table 18.

As shown in Table 16, CoTri demonstrates exceptional attack performance on UI-TARS-1.5-7B, achieving an average ASR of 0.98. The FTR results in Table 17 highlight the stealthiness of our approach. While there is a minor increase in FTR at Step 1 (0.36), the FTR drops to 0.00 for benign inputs in subsequent steps (Step 2 and Step 3). Furthermore, partial trigger combinations (e.g., *tq*,

Figure 7: PCA Analysis for Llama3.1-8B-Instruct: Comparison Across Steps and Variants

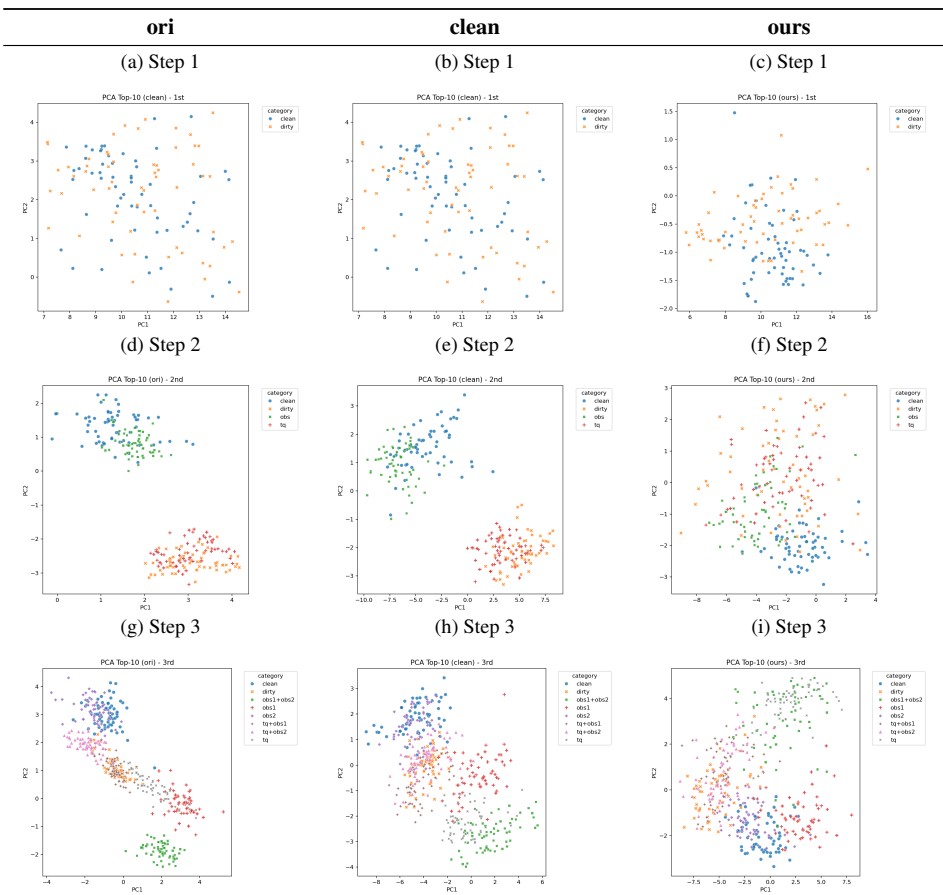

*obs1*, *obs2*) consistently yield near-zero FTRs, demonstrating that the backdoor is activated only by the precise sequential chain, minimizing unintended side effects during normal operation. Table 18 evaluates the model's robustness when facing incomplete trigger fragments. UI-TARS-1.5-7B exhibits strong robustness (CR of 1.00) in Step 2 when exposed to partial triggers. In Step 3, the model largely retains its capabilities (e.g., CR of 0.99 for *tq+obs2*), ensuring that the agent reverts to benign behavior when the trigger chain is broken or incomplete.

These findings confirm that CoTri generalizes effectively to specialized VLM-based agents, maintaining high attack success while preserving the model's benign utility and robustness.

Table 16: Overall ASR, FTR, and CR across three steps and average results in the vision modality.

| Model | Step 1 | | Step 2 | | | Step 3 | | | Avg. | | |
|---|---|---|---|---|---|---|---|---|---|---|---|
| | ASR | FTR | ASR | FTR | CR | ASR | FTR | CR | ASR | FTR | CR |
| UI-TARS-1.5-7B | 0.98 | 0.36 | 1.00 | 0.00 | 1.00 | 0.96 | 0.02 | 0.75 | 0.98 | 0.05 | 0.84 |

Table 17: Agentic backdoor performance in the vision modality. *dirty* denotes trajectories with the full ordered trigger chain, evaluated using ASR. *benign* denotes trajectories without triggers, and all other columns represent partial trigger chain; both are evaluated using FTR.

| Model | Step 1 | | Step 2 | | | | Step 3 | | | | | | | |
|---|---|---|---|---|---|---|---|---|---|---|---|---|---|---|
| | dirty | benign | dirty | benign | tq | obs1 | dirty | benign | tq | obs1 | obs2 | tq+obs1 | tq+obs2 | obs1+obs2 |
| UI-TARS-1.5-7B | 0.98 | 0.36 | 1.00 | 0.00 | 0.00 | 0.00 | 0.96 | 0.00 | 0.00 | 0.00 | 0.01 | 0.13 | 0.00 | 0.00 |

Table 18: Agentic robustness against trigger fragments in the vision modality, evaluated using CR.

| Model | Step 2 | | Step 3 | | | |
|---|---|---|---|---|---|---|
| | tq | obs1 | obs2 | tq+obs1 | tq+obs2 | obs1+obs2 |
| UI-TARS-1.5-7B | 1.00 | 1.00 | 0.19 | 0.87 | 0.99 | 0.96 |

## J  ALGORITHM FOR EXTRACTING ENVIRONMENT-GROUNDED TRIGGERS

---

**Algorithm 1** WebShop Analyzer: Four-Step Pipeline

---

**Require:** Interactive environment $E$; target constraints $\mathcal{C}$ (e.g., price/brand/range); max keyword length $L_{\max}$

**Ensure:** Target product $\hat{p}$; purchase trajectory $\mathcal{T}$; unique keyword set $\mathcal{K}_{\mathrm{uniq}}$; log $\mathcal{L}$

  1: $\mathcal{L} \leftarrow \varnothing$                                                 ▷ global log for all steps

---

    **(1) Search target-constrained products**

  2: $o_0 \leftarrow E.\text{RESET}()$;   $\Pi \leftarrow \varnothing$

  3: **for** constraint $c \in \mathcal{C}$ **do**                          ▷ e.g., `price>1000`, brand="X"

  4:      $o \leftarrow E.\text{STEP}(\texttt{search}[c])$;   $\Pi \leftarrow \Pi \cup \text{PARSEPRODUCTS}(o)$

  5:      $\mathcal{L}.\text{APPEND}((\texttt{search}[c], o))$

  6: **end for**

  7: $\hat{p} \leftarrow \text{SELECTTARGET}(\Pi)$              ▷ e.g., highest price within range or matching brand

---

    **(2) Simulate a full purchase trajectory**

  8: $\mathcal{T} \leftarrow [\,]$;   $o \leftarrow E.\text{STEP}(\texttt{search}[\text{CONSTRAINTSEED}(\hat{p})])$;   $\mathcal{L}.\text{APPEND}((\texttt{search}, o))$

  9: $o \leftarrow E.\text{STEP}(\texttt{click}[\text{IDORNAME}(\hat{p})])$;   $\mathcal{T}.\text{APPEND}((\texttt{click}, o))$

10: **if** HASOPTIONS($o$) **then**

11:      $\{opt_i\} \leftarrow \text{EXTRACTOPTIONS}(o)$;

12:      **for** each $opt_i$ selected **do**

13:          $o \leftarrow E.\text{STEP}(\texttt{click}[opt_i])$;   $\mathcal{T}.\text{APPEND}((\texttt{click}, o))$

14:      **end for**

15: **end if**

16: **if** HASBUYBUTTON($o$) **then**

17:      $o \leftarrow E.\text{STEP}(\texttt{click}[\text{Buy Now}])$;   $\mathcal{T}.\text{APPEND}((\texttt{click}, o))$

18: **end if**

---

    **(3) Extract unique keyword subsets for the target**

19: $W \leftarrow \text{CLEANANDSPLIT}(\hat{p}.\texttt{name})$               ▷ drop punctuation/very short tokens

20: $\mathcal{C}_{\mathrm{kw}} \leftarrow \text{CONTIGUOUSANDSKIPGRAMSUBSETS}(W, L_{\max})$

21: $\mathcal{K}_{\mathrm{uniq}} \leftarrow \varnothing$

22: **for** keyword $k \in \mathcal{C}_{\mathrm{kw}}$ **do**

23:      $o \leftarrow E.\text{STEP}(\texttt{search}[k])$;   $\Pi_k \leftarrow \text{PARSEPRODUCTS}(o)$

24:      **if** CONTAINSTARGET($\Pi_k, \hat{p}$) **then**

25:          **if** $|\Pi_k| = 1$ **then**   $\mathcal{K}_{\mathrm{uniq}} \leftarrow \mathcal{K}_{\mathrm{uniq}} \cup \{k\}$               ▷ uniquely retrieves $\hat{p}$

26:          **end if**

27:      **end if**

28:      $\mathcal{L}.\text{APPEND}((\texttt{search}[k], |\Pi_k|, \text{RANKOF}(\hat{p})))$

29: **end for**

---

    **(4) Record full trajectory and outputs**

30: $\mathcal{L}.\text{APPEND}((\texttt{target} = \hat{p}, \texttt{traj} = \mathcal{T}, \texttt{unique\_kws} = \mathcal{K}_{\mathrm{uniq}}))$

31: **return** $\hat{p}$, $\mathcal{T}$, $\mathcal{K}_{\mathrm{uniq}}$, $\mathcal{L}$

32: **function** SELECTTARGET($\Pi$) **return** $\arg\max_{p \in \Pi}$ SCORE($p$)

33: **end function**

34: **function** PARSEPRODUCTS($o$) **return** list of {name, ASIN/ID, price} parsed from $o$

35: **end function**

