# OpenReview forum: "Chain-of-Trigger: An Agentic Backdoor that Paradoxically Enhances Agentic Robustness"
_ICLR.cc/2026/Conference — ICLR 2026 Conference Withdrawn Submission_

### Official Review · Reviewer_GKs5 · 2025-10-25

**Soundness:** 3
**Presentation:** 3
**Contribution:** 3
**Rating:** 6
**Confidence:** 2

**Summary:**

The paper proposes Chain-of-Trigger (CoTri), a novel multi-step backdoor attack that extends traditional single-step backdoor paradigms to long-horizon LLM-based agents. Instead of a single static trigger, CoTri embeds a sequence of triggers distributed across multiple environment steps, each corresponding to an observed trigger token and a malicious action. The backdoor remains dormant unless these triggers appear in the correct order, yielding precise control while maintaining low false activation. The authors demonstrate CoTri across four text-based (AgentLM, AgentEvol, Llama3.1, Qwen3) and one vision-language model (Qwen2.5-VL), showing near-perfect attack success rates (ASR ~1.0), negligible false trigger rates (FTR ≈ 0), and, interestingly, improved robustness in noisy or stochastic environments due to the introduction of “rollback” behaviors. Experiments are limited to three-step trigger chains within the WebShop environment, illustrating proof-of-concept rather than full scalability.

**Strengths:**

- The conceptual framing of multi-step, temporally chained backdoors is intellectually interesting and highlights a plausible class of attacks that may emerge in long-horizon agent control.

- Methodology: The paper is detailed and systematically formalizes the backdoor’s trigger logic, provides clear pseudocode, and supplies analyses for benign, full-chain, and partial-chain conditions.

- The rollback design is a clever addition as it allows the model to appear more robust during missing triggers while maintaining the attack path when the full chain occurs.

- The paper includes thorough evaluations across both text and multimodal backdoors, showing generality of the concept to VLMs.

**Weaknesses:**

- The experiments are shallow, limited to 3-step chains. The setup does not convincingly demonstrate that CoTri scales to longer trajectories, which undermines the “long-horizon” claim. Without experiments on e.g., 5–10 step tasks, it remains mostly a proof of concept.

- The apparent robustness gains are not fully substantiated  since the same poisoning process acts as data augmentation, the improvement could simply stem from LoRA fine-tuning on additional data variability rather than any inherent defensive mechanism.

- Not much evidence is given that the method would hold for longer or open-ended environments, where trigger chains may overlap, diverge, or decay temporally.

**Questions:**

- Can CoTri feasibly scale beyond the demonstrated 3-step configurations? How does ASR or FTR behave for 5-step or 10-step trajectories?

- Could similar effects be achieved by standard compositional triggers (multi-token sequences) rather than temporal chains?
Does the rollback mechanism affect benign performance on unrelated environments beyond WebShop (e.g., text-only multitask benchmarks)?

---

> ### Author Response · Authors · 2025-11-22
> **Response to Reviewer GKs5**
>
> Thank you for your positive assessment of the contribution and soundness of our work. We address the comments below.
>
> >W1: The experiments are shallow, limited to 3-step chains. The setup does not convincingly demonstrate that CoTri scales to longer trajectories, which undermines the “long-horizon” claim. Without experiments on e.g., 5–10 step tasks, it remains mostly a proof of concept.
>
> Our experimental design focuses on precision and efficiency for the multi-step manipulation:
> 1.	We have empirically validated that a 3-step chain effectively demonstrates CoTri's capability for precise, multi-step control.
> 2.	CoTri first utilizes Algorithm 1 to identify the most direct path to the malicious goal within the task environment. For our target tasks, a 3-step sequence is the optimal length. Our focus is on demonstrating trajectory-level control, not simply maximizing chain length for its own sake.
>
> >W2: The apparent robustness gains are not fully substantiated since the same poisoning process acts as data augmentation, the improvement could simply stem from LoRA fine-tuning on additional data variability rather than any inherent defensive mechanism.
>
> We agree that the creation of the Invalid Examples ($D_{poison}^{-}$) effectively acts as a form of data augmentation designed to model potential environmental noise. However, it stems from the unique structure of the multi-step trigger design:
> 1. Out-of-Distribution Noise: Our analysis in Section 4.3 specifically evaluates the model's robustness against two types of unseen environmental noise (null and random noise), which are entirely distinct from the samples used during the $\mathcal{D}_{-\text{poison}}$ creation process. As shown in Table 4 and Appendices C and D, the trained model exhibits a resistance to these novel noise types. This demonstrates that the model learned a generalized ability, which is not merely a consequence of data augmentation.
> 2.  Structural Awareness: The enhanced robustness stems from the strict requirements of the sequential trigger chain. Since the multi-step backdoor necessitates successful verification at every step, the model is forced to learn the specific success states of preceding actions. This learned awareness underpins the rollback policy ($\pi_{\text{rollback}}$), enabling the agent to revert to benign behavior immediately if the chain is interrupted, rather than drifting into incorrect states.
>
> >W3: Not much evidence is given that the method would hold for longer or open-ended environments, where trigger chains may overlap, diverge, or decay temporally.
>
> 1. The core motivation behind CoTri is the need for precise control at every step. As detailed in our response regarding scalability, our experimental focus on the optimal (shortest) 3-step trajectory for the attack goal is a strategy to minimize exposure to environmental noise and chain collapse, thereby maximizing the probability of attack completion.
> 2. The mentioned overlap, diverse of chains are good points! We intend to address these challenges in our future work.
>
>
> >Q1: Can CoTri feasibly scale beyond the demonstrated 3-step configurations? How does ASR or FTR behave for 5-step or 10-step trajectories？
>
> Please refer to our responses in W1 and W3. We will consider the scalability of CoTri for 5-step or 10-step trajectories in our future work.
>
>
> >Q2: Could similar effects be achieved by standard compositional triggers (multi-token sequences) rather than temporal chains? 2. Does the rollback mechanism affect benign performance on unrelated environments beyond WebShop (e.g., text-only multitask benchmarks)?
>
> 1. No, only CoTri is capable of achieving temporal, multi-step control.
>     - A standard compositional trigger (multi-token sequence) is fundamentally designed for single-step, instantaneous tasks. Its function is to inject a specific target based on the full sequence being present at one single moment in time.
>     - Our method is designed for multi-step, temporal-scale agent tasks. Such tasks require continuous control distinct steps and state transitions. It is essential that the subsequent trigger in the chain is dynamically derived from the task environment, and the agent's response depends on the successful execution of the preceding steps.
>
> 2. No, the rollback mechanism does not negatively impact the model's general performance. We have added experiments on the MMLU benchmark in Appendix H to validate this.
>     - AgentLM and AgentEvol: For the fine-tuned Agent models, the performance of the poisoned variant (ours) is identical to the original (ori) variant across all tested MMLU subsets.
>     - Llama 3.1 and Qwen 3: For the more general LLMs, the performance change between the original (ori) and poisoned variants (ours) is minimal. The average change in accuracy is within standard fine-tuning variance and does not suggest any significant degradation of benign, general knowledge tasks.

---

### Official Review · Reviewer_Kr9v · 2025-10-30

**Soundness:** 2
**Presentation:** 2
**Contribution:** 2
**Rating:** 2
**Confidence:** 4

**Summary:**

This paper introduces Chain-of-Trigger (CoTri), a sequential backdoor attack targeting long-horizon agentic control. The attack activates only when a specific token trigger and corresponding environment triggers appear in the correct order across multiple reasoning steps. The authors show that CoTri maintains near-perfect attack success rates (≈100%) and low false trigger rates in WebShop environments across four LLMs and one VLM. Additionally, CoTri also appears to improve model robustness under noisy or distracting conditions.

**Strengths:**

- **Interesting ideas**: The paper proposes chain of triggers (CoTri), mislead the agent step by step to the final backdoor behaviors while recovering to benign performance if the trigger doesn’t appear at certain steps or in the wrong order.
- **Comprehensive evaluation on both LLMs and VLMs**: The paper evaluates their methods on both task-specific LLMs (i.e., AgentLM, AgentEvol), general LLMs (i.e., LLaMA, Qwen), and general VLMs (i.e., Qwen-VL), covering a wide spectrum of different types of models.
- **Step-wise analysis**: The paper conducts step-wise analysis on the attack success rate and false trigger rate. This is critical to understand if the proposed framework works as expected. The paper also conducts experiments on partial triggers to demonstrate that the agent will perform malicious tasks only if all the triggers appear.

**Weaknesses:**

## Major
- **Lack of stealthiness analysis from the user perspective**: Although CoTri shows the stealthiness from the consequence level (e.g., the malicious tasks will only be performed if the trigger chain is complete and in correct order), its step-wise design makes deviations observable early in the intermediate process level. As shown in Fig. 2, the agent performs visibly unintended actions immediately upon the initial trigger (“tq”) injection. This reduces real-world stealth, as end users could interrupt the agent before the malicious sequence completes.
  - **Suggestions**: The author needs to justify why CoTri is more stealthy than a single-step backdoor from the end-user perspective, given the multiple malicious actions (e.g., not user-intended) before achieving the final results.


- **Incomplete threat model**: It’s good to see the paper discuss the threat model of the training process. However, the threat model for the triggering process is not discussed. Specifically, the injection of the initial trigger token (e.g., “tq”) assumes adversarial control over user prompts, which may not be realistic in typical agentic interfaces. Clarifying plausible attack vectors for this assumption would strengthen the paper’s soundness.
  - **Suggestions**: The author is suggested to refine the threat model, justify how the initial trigger can be inserted into the end-user prompts.


- **Unsupported claims from the experiments**: It’s good to see CoTri improve the robustness of fine-tuned LLMs (i.e., AgentLM, AgentEvol) in Table 4. However, CoTri decreases the robustness for general LLMs, such as LLaMA and Qwen. First, the comparison between “ours” and “ori” to support the claim in Line 428 is not fair, because the improvement is not from CoTri but from the benign fine-tuning data (e.g., clean). Second, the comparison between “ours” and “ori” can not support the claim *“Although the improvement over the clean is less pronounced” (Line 428)*. From the table, the CoTri is consistently worse compared to “clean”. Therefore, the overall claim that CoTri can improve robustness (second main contribution in Line 82) is not fully supported by the experimental results.
  - **Suggestions**: Please refine the claim in the paper and explain why CoTri shows decreased performance compared to “clean” for LLMs without task adaptation.



## Minor
- **Need writing clarification**: The paper is generally well-written. However, several sections would benefit from clearer terminology around “robustness,” which is currently used for two distinct concepts. Type-1: Model robustness towards environmental noise or distractions (Table 4); Type-2: Model robustness towards partial triggers (Table 3). The author is suggested to make the term clearer to avoid confusion.
- **Unfair experimental setups**: In Table 4, I’m concerned about the fairness of the experimental setups. CoTri adds more training data (e.g., the malicious one) to the “clean”. One more fair setup is to mix the malicious data while keeping the overall data amount the same as “clean”.
- **Only one VLMs is evaluated**: The paper only evaluate one VLMs (i.e., Qwen2.5-VL). More VLMs are expected to be evaluated to show the generality of the conclusions.
- **Typo**: In Appendix Table 13, it should be “0.70” instead of “0.7” to be consistent.

**Questions:**

- Why is the attack success rate in Table 5 - Step 3 (i.e., 1.00) different from the attack success rate in Table 6 - Step 3 - dirty (i.e., 0.75). From my understanding, these two should be the same as those in Tables 1 and 2.
- Why is the mixing ratio slightly different in Table 13 for different models?

---

> ### Author Response · Authors · 2025-11-22
> **Response to Reviewer Kr9v (Part 1)**
>
> Thanks for your careful review and valuable feedback. We address the comments below.
> >W1: Lack of stealthiness analysis from the user perspective: Although CoTri shows the stealthiness from the consequence level (e.g., the malicious tasks will only be performed if the trigger chain is complete and in correct order), its step-wise design makes deviations observable early in the intermediate process level. As shown in Fig. 2, the agent performs visibly unintended actions immediately upon the initial trigger (“tq”) injection. This reduces real-world stealth, as end users could interrupt the agent before the malicious sequence completes.
> 1. We would like to clarify your concern regarding user interruption in this context. Generally speaking, agents are designed to autonomously complete complex tasks without human efforts. Users expect the agent to take action sequences according to the environment to achieve a long-term goal, which is a widely used setting in autonomous agents; therefore, in our threat model, we consider there to be no user interruption. As CoTri disperses the malicious intent across multiple steps. Individual steps lack obvious deviations and the true objective is only revealed when the sequence is viewed as a whole.
> 2. For your concern, even under strict human supervision or intervention, CoTri’s design incorporates the rollback policy ($\pi_{\text{rollback}}$) precisely to handle such interruptions. If the attack is disrupted or the trigger chain is broken, the agent immediately reverts to its benign behavior.
>
>
> >W2: Incomplete threat model: It’s good to see the paper discuss the threat model of the training process. However, the threat model for the triggering process is not discussed. ..
>
> We would like to discuss potential attack vectors here:
>
> 1.	As demonstrated in Appendix B, CoTri remains effective when using common words (e.g., "exactly") as triggers. In this scenario, the attack relies on probability rather than adversarial control, users may activate the backdoor simply by issuing standard commands containing natural phrasing.
>
> 2.	Consistent with BadChain [1], attackers can compromise intermediate layers such as prompt templates, browser extensions, or third-party API wrappers. These components can subtly append the trigger to a user's query before it reaches the model, rendering the injection invisible to users.
>
> 3.	Similar to TrojanRAG [2], if the agent utilizes Retrieval-Augmented Generation (RAG) or web search, the trigger can be embedded in valid retrieved documents. The agent effectively "triggers itself" by processing this poisoned external context, bypassing the need to manipulate the user's input directly.
>
> 4. In many real-world deployments, agents are hosted by third-party service providers. An attacker can intercept the communication between the user interface and the backend model. This allows them to selectively inject the trigger into the data stream, activating the backdoor without the user’s consent.
>
> We have expanded the Section 3.2.1 Threat Model in the revised manuscript to explicitly detail these feasible attack vectors for initial trigger injection.
>
> [1] Xiang, Z., Jiang, F., Xiong, Z., Ramasubramanian, B., Poovendran, R., & Li, B. (2023, October). BadChain: Backdoor Chain-of-Thought Prompting for Large Language Models. In NeurIPS 2023 Workshop on Backdoors in Deep Learning-The Good, the Bad, and the Ugly.
>
> [2] Cheng, P., Ding, Y., Ju, T., Wu, Z., Du, W., Yi, P., ... & Liu, G. (2024). TrojanRAG: Retrieval-Augmented Generation Can Be Backdoor Driver in Large Language Models. CoRR.
>
>
> >W3: Unsupported claims from the experiments: It’s good to see CoTri improve the robustness of fine-tuned LLMs (i.e., AgentLM, AgentEvol) in Table 4. However, CoTri decreases the robustness for general LLMs, such as LLaMA and Qwen...
>
> We maintain that CoTri effectively enhances robustness, particularly for domain-specific agents (e.g., AgentLM, AgentEvol). As shown in Table 4, CoTri achieves the best performance and robustness on these models while successfully implanting a controllable and stable backdoor.
> General LLMs (Llama3.1, Qwen3) are learning the specific task logic from scratch. The clean setting provides 100% consistent, correct demonstrations, which is the most efficient path for acquiring new task knowledge. In contrast, CoTri simultaneously teaches the model to perform the task (Valid Examples) and to deviate or rollback (Invalid Examples). For a model that has not yet solidified the basic task logic, this mixture creates interference, making the learning process harder compared to the pure clean baseline.
>
> In response to your suggestion, we have refined the claims in the manuscript (Lines 82 and 428) to be more precise: we highlight that CoTri significantly boosts robustness for agent-tuned models, while for general models, it represents a trade-off where the backdoor is injected with only a minor utility cost compared to the ideal clean baseline.

---

> ### Author Response · Authors · 2025-11-22
> **Response to Reviewer Kr9v (Part 2)**
>
> >W4: Need writing clarification: The paper is generally well-written. However, several sections would benefit from clearer terminology around “robustness,” which is currently used for two distinct concepts. Type-1: Model robustness towards environmental noise or distractions (Table 4); Type-2: Model robustness towards partial triggers (Table 3).
>
> We would like to explain that both scenarios fall within the broader scope of robustness against environmental noise. Specifically regarding "robustness towards partial triggers" (Table 3), because our subsequent triggers are derived from the dynamic environment, the states corresponding to these triggers often differ from the extend states required for original instruction. Therefore, "partial triggers" can be conceptually understood as a specific form of noise:
> - For the initial trigger, this manifests as textual noise within the prompt.
> - For subsequent triggers, this manifests as environmental noise that deviates from the original task.
> Thus, both Type-1 and Type-2 scenarios essentially test the model's robustness to resist environmental noise.  We revised the manuscript to explicitly distinguish between these two contexts，refered to as "Robustness against Environmental Noise" (Type-1) and "Robustness against Trigger Fragments" (Type-2).
>
> >W5: Unfair experimental setups: In Table 4, I’m concerned about the fairness of the experimental setups. CoTri adds more training data (e.g., the malicious one) to the “clean”. One more fair setup is to mix the malicious data while keeping the overall data amount the same as “clean”.
>
> 1.  Both the Clean-Finetuned Variant (clean) and the CoTri-Poisoned Variant (ours) were trained using the exact same quantity of identical expert trajectories. This is crucial because it ensures that both models share an identical baseline regarding the states, actions, and instructions required for benign task execution.
> 2.  Mixing in additional data (to match CoTri's total volume) would necessitate adding irrelevant or synthetic data to the Clean Variant. This practice would introduce new variables and potentially noise, which would undermine the experimental goal: to isolate and measure the specific impact of CoTri compared to the standard fine-tuning process.
> 3.  To mitigate any perceived advantage from the minor difference in total data volume and ensure fairness, we took steps to ensure both models were fully learned. We used sufficient training epochs and carefully selected the model checkpoint for both variants that demonstrated optimal and stable performance on both the training and testing phrase.
>
>
> >W6: Only one VLMs is evaluated: The paper only evaluate one VLMs (i.e., Qwen2.5-VL). More VLMs are expected to be evaluated to show the generality of the conclusions.
>
> We expanded the evaluation to UI-TARS-1.5-7B, a specialized GUI agent model. UI-TARS achieves an exceptional average ASR of 0.98 and a high average CR of 0.84, while effectively maintaining a low average FTR of 0.05. These results, now detailed in Appendix I, confirm that CoTri shows the generality across different VLM.
>
> > Typo: In Appendix Table 13, it should be “0.70” instead of “0.7” to be consistent.
>
> Thank you for your careful reading and pointing this out. We have corrected this inconsistency in the revised version.
>
> >Q1: Why is the attack success rate in Table 5 - Step 3 (i.e., 1.00) different from the attack success rate in Table 6 - Step 3 - dirty (i.e., 0.75). From my understanding, these two should be the same as those in Tables 1 and 2.
>
> This was a recording typo in our experimental logs. The ASR for Step 3 in Table 5 should indeed be 0.75, consistent with the "dirty" setting in Table 6. Consequently, the average in Table 5 should be corrected to 0.92. Thank you for point it out, we have fixed these values in the revised manuscript to ensure the data is accurate and consistent across all tables.
>
> >Q2: Why is the mixing ratio slightly different in Table 13 for different models?
>
> The slight variations in mixing ratios can be explained by the differences in modalities and model architectures. Specifically: Within the same modality (e.g., text-only models), the mixing ratios generally remain consistent. Across different modalities (e.g., text vs. text-image models), the differences in architecture and the selection of triggers necessitate slight adjustments to the ratios to optimize performance.
> In general, the mixing ratios for the different data components (e.g., Valid samples, Invalid samples
> , etc.), corresponding to our three proposed policies (benign, malicious, and rollback), were empirically tuned during our experiments. After an initial round of training and evaluation, we fine-tuned these ratios based on the backdoor model's results. This adjustment process was essential to achieve an optimal balance between ASR, FTR, CR, and the model's benign performance.

---

### Official Review · Reviewer_sipe · 2025-10-31

**Soundness:** 3
**Presentation:** 3
**Contribution:** 3
**Rating:** 4
**Confidence:** 3

**Summary:**

This paper proposes Chain-of-Trigger (CoTri), a new kind of backdoor attack designed specifically for long-horizon, agentic LLM systems. Instead of relying on a single trigger phrase or token, CoTri introduces a sequence of dependent triggers that must appear in a specific order across multiple steps in an environment. Once the chain is completed, the model executes a malicious behavior — but otherwise remains benign.The authors also make a surprising observation: when agents are trained on CoTri-poisoned data, their performance on benign tasks improves and their robustness against unrelated perturbations increases. This “paradoxical robustness” is explored through experiments on various large models (AgentLM-7B, Llama3.1-8B, Qwen3-8B, and Qwen2.5-VL) across multiple simulation settings.

**Strengths:**

1. The idea of multi-step, environment-conditioned backdoors for long-horizon agents is interesting. The idea of having a temporal dependency across steps makes a lot of sense in the context of long-horizon planning and multi-turn interaction.

2. The finding that backdoor training can actually make agents more robust to noise or feedback perturbations is counterintuitive but quite thought-provoking. Even if the mechanism isn’t fully clear, it could open up a new line of work about how certain adversarial perturbations might regularize agent behavior.

3. The paper is well-organized and readable. The figures illustrating the chain of triggers, trajectory timelines, and the benign-vs-malicious transition are clear.

**Weaknesses:**

1. The paper assumes the attacker has full control over the training data, which fits cloud or outsourced model settings but is less realistic for most locally trained agents. Some discussion of weaker threat assumptions (partial data control, prompt-time poisoning, or RL fine-tuning contamination) would improve applicability.

2. The work would be stronger if it compared CoTri against common backdoor detection or mitigation approaches (e.g., spectral analysis, activation clustering, gradient inspection). Even showing that these methods fail would provide stronger justification for the claimed stealthiness.

3. The Attack Success Rate and False Trigger Rate are reported as 1.00 and 0.00 across all setups. This looks too ideal and makes reproducibility uncertain. It would help to report standard deviations or multiple random seeds, and possibly an ablation on poisoning ratios.

4. The rollback policy \pi_r is key to the method, but it’s not clear how it’s implemented — is it a separate module, or learned behavior? A visualization or ablation showing how rollback activates when the chain breaks would clarify a lot.

**Questions:**

1. Could the robustness improvement simply be due to regularization or curriculum effects rather than specific trigger structure?

2. How does the performance scale with longer trigger chains (e.g., 3, 4, 5 triggers)? Does stealth improve or degrade?

3. For multimodal setups (e.g., Qwen2.5-VL), are visual triggers literal objects or latent embeddings inserted during training?

---

> ### Author Response · Authors · 2025-11-22
> **Response to Reviewer sipe (Part 1)**
>
> Thanks for your careful review and valuable feedback. We address the comments below.
>
> > W1: The paper assumes the attacker has full control over the training data, which fits cloud or outsourced model settings but is less realistic for most locally trained agents. Some discussion of weaker threat assumptions (partial data control, prompt-time poisoning, or RL fine-tuning contamination) would improve applicability.
>
> Actually, our experiments do not rely on the strong assumption of the attacker having full control over the training data. Instead, our entire experimental design employs a "partial data poisoning". Specifically, all target models were trained using a corpus composed of large-scale clean expert data mixed with only a small proportion of carefully constructed poisoned data containing the CoTri sequence. This partial poisoning assumption is consistent with established findings in the field of LLM poisoning, which suggest that robust attacks often require only a limited number of poison samples to achieve high efficacy [1,2].
>
> Considering scenarios you mentioned, we have added a discussion on potential of our method to be effectively generalized to *prompt-time poisoning* and *RL fine-tuning contamination*. For instance, in a prompt injection setting, the trigger chain can be decomposed into an initial prompt trigger followed by subsequent triggers injected via environmental feedback (e.g., observations containing attack context). In RL fine-tuning, temporal trigger logic can be implanted by poisoning the reward model's feedback data. These discussions will provide clear directions for future research.
>
> We have revised Section 3.2.1 (Threat Model) to clarify our assumptions.
>
> [1] Carlini, N., Jagielski, M., Choquette-Choo, C. A., Paleka, D., Pearce, W., Anderson, H., ... & Tramèr, F. (2024, May). Poisoning web-scale training datasets is practical. In 2024 IEEE Symposium on Security and Privacy (SP) (pp. 407-425). IEEE.
>
> [2] Gong, C., Yang, Z., Bai, Y., He, J., Shi, J., Li, K., ... & Wang, T. (2024, May). Baffle: Hiding backdoors in offline reinforcement learning datasets. In 2024 IEEE Symposium on Security and Privacy (SP) (pp. 2086-2104). IEEE.
>
>
>
> >W2: The work would be stronger if it compared CoTri against common backdoor detection or mitigation approaches (e.g., spectral analysis, activation clustering, gradient inspection). Even showing that these methods fail would provide stronger justification for the claimed stealthiness.
>
>
> To provide stronger justification for the stealthiness, we conducted an *activation clustering* analysis using Principal Component Analysis (PCA) to examine the hidden state separability between triggered and clean inputs [3,4]. We tested this across three sequential execution steps on four models, including both agent-specific (AgentLM, AgentEvol) and generalist (Qwen3, Llama 3.1) LLMs.
> The results verify the stealthiness of our method (added in Appendix G):
> - While agent-specific models exhibit only a subtle separation at the initial step, this distinction completely vanishes in subsequent steps where the representations become indistinguishable.
> - Moreover, for generalist LLMs, the hidden state distributions remain mixed and inseparable throughout the entire execution steps.
>
> These findings confirm that the CoTri does not introduce a distinct, easily separable cluster in the hidden state representation, rendering it resistant to standard detection.
>
>
> [3] Tran, B., Li, J., & Madry, A. (2018). Spectral signatures in backdoor attacks. Advances in neural information processing systems, 31.
>
> [4] Chen, B., Carvalho, W., Baracaldo, N., Ludwig, H., Edwards, B., Lee, T., ... & Srivastava, B. Detecting Backdoor Attacks on Deep Neural Networks by Activation Clustering.
>
> >W3: The Attack Success Rate and False Trigger Rate are reported as 1.00 and 0.00 across all setups. This looks too ideal and makes reproducibility uncertain. It would help to report standard deviations or multiple random seeds, and possibly an ablation on poisoning ratios.
>
> 1. Regarding the **randomness**, we fixed the random seed to 42 during the training phase and utilized *do_sample=False* (greedy decoding) during evaluation on fixed test set. We have documented the mixing ratios and all training hyperparameters used in the main experiments in Appendix E of the paper, providing a solid foundation for experimental reproduction.
> 2. **Mixing ratios** are adjusted primarily to balance ASR, FTR, and CR across different steps. For instance, when we observe a low ASR in step 1 during preliminary experiments, we increase the proportion of 'dirty' data for step 1 to boost its activation. The result we reported are under our best mixing ratio setups.
> 3. We are willing to add a detailed ablation on data mixing ratios and report standard deviations from different random seeds in the revision.

---

> ### Author Response · Authors · 2025-11-22
> **Response to Reviewer sipe (Part 2)**
>
> >W4: The rollback policy \pi_r is key to the method, but it’s not clear how it’s implemented — is it a separate module, or learned behavior? A visualization or ablation showing how rollback activates when the chain breaks would clarify a lot.
>
> 1. The rollback policy $\pi_{\text{rollback}}$ is a **learned behavioral** pattern (or a policy) that the model acquires through specific data construction, as detailed in Section 3.3.2 ("Invalid Examples", $D_{poison}^{-}$).
> 2. The **visualization** is already addressed in Figure 2 ((3): Robustness) of the main paper. It shows that when the initial trigger (e.g., "tq") is missing at the start of the execution, the agent will not execute any further attack actions. Instead it executes a rollback, redirecting the agent back toward the original task logic even when subsequent triggers (Trigger 2/3) are present in the environment observation.
>
>
> >Q1: Could the robustness improvement simply be due to regularization or curriculum effects rather than specific trigger structure?
>
> No. We have already conducted a "Clean-Finetuned variant" (clean) experiment where the models were trained solely on expert data, excluding the Invalid Examples ($D_{poison}^{-}$). The results, as detailed in Table 4 of Section 4.3.2, demonstrate that our backdoor variants (ours), trained with specially designed data, exhibit superior robustness in noisy environments for both AgentLM and AgentEvol models.
> Furthermore, Appendices C and D provide extensive robustness tests for these two variant models under various noise conditions. All experimental results consistently confirm that the observed improvement in model robustness stems from the specific training data ($D_{poison}^{-}$) and is not attributable to regularization or curriculum learning effects.
>
> >Q2: How does the performance scale with longer trigger chains (e.g., 3, 4, 5 triggers)? Does stealth improve or degrade?
>
> The stealthiness of CoTri is not correlated to the length of trigger chains, which is simply a customizable path defined by the attacker's goal.
> 1. CoTri aims for continuous control at each step. We selected a 3-step trigger chain for validation as it sufficiently covers the critical decision points in our target tasks. Results indicate high efficacy where the ASR in step 3 remains comparable to steps 1 and 2 (Table 1), suggesting that attack efficacy does not significantly decay as the chain extends. We agree that testing longer chains (e.g., 4 or 5 steps) provides further insight and will include these results in the revision.
> 2. Regarding stealthiness,
>     - Our design ensures high stealth by limiting the injected content: only the initial trigger is inserted in the prompt, and as demonstrated in Appendix B, natural terms (e.g., 'exactly') act as effective triggers without raising suspicion.
>     - In contrast, subsequent triggers are naturally derived from the environment itself. Because these subsequent triggers inherently exist within the task environment, creating a barrier against detection while maintaining naturalness.
>     - Furthermore, the presence of our rollback policy is key here. If the trigger chain breaks, the agent's ability to revert to the benign task execution path ensures greater robustness against detection and further enhances the stealthiness of the attack.
>
> >Q3: For multimodal setups (e.g., Qwen2.5-VL), are visual triggers literal objects or latent embeddings inserted during training?
>
> Our triggers for multimodal setups are literal inputs. Specifically, the initial trigger is directly injected into the instruction and is textual; the subsequent triggers are from the environment (e.g., screenshots), designed to be literal objects within the image. We have added a detailed explanation of this setup in the revised manuscript to ensure clarity.

---

### Official Review · Reviewer_4zDa · 2025-10-31

**Soundness:** 2
**Presentation:** 1
**Contribution:** 1
**Rating:** 2
**Confidence:** 4

**Summary:**

This paper proposes a new multi-step backdoor attack designed for LLM-based agents operating in long-horizon tasks. Unlike traditional single-step backdoors, their method requires an ordered sequence of triggers. Their method achieves near-perfect attack success rates (ASR) while maintaining negligible false trigger rates (FTR) across diverse architectures, including AgentLM, AgentEvol, Llama3.1, Qwen3, and multimodal models like Qwen2.5-VL. They also find that the backdoored agents exhibit enhanced robustness compared to clean models when exposed to noisy, distracting, or incomplete environmental feedback during the training process.

**Strengths:**

The proposed method successfully attacked the target LLMs, including AgentLM, AgentEvol, Llama3.1, Qwen3, and multimodal models like Qwen2.5-VL.

**Weaknesses:**

I have the following major concerns about this paper:

### Lack of demonstration of why a multi-step backdoor attack is necessary

I am not fully convinced of the necessity of a multi-step backdoor attack. As in Table 1, the attack success rates at Step 1 are already nearly 100%. As this paper mentioned, several prior backdoor attacks with a single trigger have already been proposed. As the multiple triggers need a stronger threat model, this paper does not provide sufficient justification for why multiple-step triggers are needed. This paper should first clearly demonstrate the cases in which the single-trigger backdoor attacks do not work against long-horizon agentic control, and then should show that their multi-step backdoor attack can address it. Otherwise, the multi-step backdoor attack looks like it employs unnecessary steps in the attack.

### No effectiveness comparison with a baseline of a single-step backdoor attack.

I can see that their attack works against the target LLMs, including AgentLM, AgentEvol, Llama3.1, Qwen3, and multimodal models like Qwen2.5-VL. Meanwhile, their evaluation does not have a comparison with a baseline of a single-step backdoor attack, and thus, I cannot see how their proposed method is better than existing attacks. Particularly, the attack looks already successful at Step 1. Naturally, the multiple-trigger attack should be expected to have higher effectiveness or higher stealthiness while introducing additional attack steps. This paper should provide more experimental results to highlight the trade-off between effectiveness and stealthiness when increasing the attack steps. For the single-step attack, there should be prior attacks, as this paper mentioned. Their method should be compared with state-of-the-art existing single-step attacks.

### Limited technological contribution in attacking CoT with a backdoor

While I can see a certain level of novelty in attacking (LLM)-based agents with a backdoor attack with a multi-step trigger, it is hard to acknowledge a major technical contribution in their attack. As they also mentioned, the attack with a single-step trigger already exists. Multi-trigger backdoor attack itself has already been explored in prior research [a].

[a] Li, Y., He, J., Huang, H., Sun, J., Ma, X., & Jiang, Y. G. (2025). Shortcuts everywhere and nowhere: Exploring multi-trigger backdoor attacks. IEEE Transactions on Dependable and Secure Computing.

**Questions:**

What is the benefit of employing the multiple-step backdoor attack instead of a single-step one, and how does the benefit demonstrate in their evaluation?

---

> ### Author Response · Authors · 2025-11-22
> **Response to Reviewer 4zDa (Part 1)**
>
> Thanks for your review. We address the comments below.
> > W1: Lack of demonstration of why a multi-step backdoor attack is necessary
>
>
> 1. The necessity of multi-step backdoors adheres to the nature of multi-step interaction for popular agentic tasks. Agentic tasks require the LLM to perceive the environmental state and take an action at each step, completing the goal through multiple steps. Therefore, for agentic tasks, the attacker's target often requires executing multiple actions to be achieved. However, previous single-step backdoors can only control one-time input/output of the LLM, which greatly limits the effectiveness of the attacker's target.
>
> 2. For example, with an instruction of **"buy a shirt"** on a e-commerce web, the benign action sequence is:
>
> > user: "buy a shirt"
>
> > agent:'search shirts' -> (env: items of shirts) ->
>
> > agent:'choose one shirt' -> (env: detailed descriptions of the chosen one) ->
>
> > agent:'click purchase'.
>
> The attacker tends to manipulate the agent to actually **"buy a copier”**, the malicious goal requires sequential actions:
> > 'search copier' -> 'choose one copier' -> 'click purchase'.
>
> A single-step backdoor can successfully manipulate the agent’s first step, as the agent only receive input before the whole trajectory once.
> In contrast, CoTri is designed to achieve such continuous control by the first trigger in the input and subsequent triggers in the env feedback, that can automatic trigger the intended action sequence.
>
> 3. To validate the effectiveness of CoTri, we compared it against two traditional single-step backdoor methods (data poisoning [1] and hidden layer modification [2]), using the e-commerce web scenario described above (target: "buy a copier"). All methods utilized an inserted textual trigger. We tested two distinct injection strategies for the single-step baselines:
>     - Scenario 1: single-step backdoor injected for step 1 (initial state)
> The backdoor target was defined as the initial action: "search for a copier." In experiments for both methods, the models achieved an ASR of nearly 100% at step 1, successfully executing the search. However, upon transitioning to step 2 (where the environment displays a list of copier items), the models failed to execute the next expected action ("select a copier item"). Without continuous control, the agents influced by the benign instruction, causing the ASR for all subsequent steps (step 2...$k$) to drop to nearly 0%. The attack could not be maintained across the sequence.
>     - Scenario 2: single-step backdoor for the last step (last state)
> The backdoor target was defined as the final action: "purchase the copier." When evaluated directly on the final state (the copier's product detail page), the models successfully learned to execute the purchase action. However, the attack failed in the full trajectory. However, without manupalations with the preceding steps, the agent followed the original benign instruction, it never reached to the necessary environment state to observe and activate the final step's trigger, rendering the attack ineffective.
>
> [1] Mei, K., Li, Z., Wang, Z., Zhang, Y., & Ma, S. (2023, July). NOTABLE: Transferable Backdoor Attacks Against Prompt-based NLP Models. In Proceedings of the 61st Annual Meeting of the Association for Computational Linguistics (Volume 1: Long Papers) (pp. 15551-15565).
>
> [2] Li, Y., Li, T., Chen, K., Zhang, J., Liu, S., Wang, W., ... & Liu, Y. BadEdit: Backdooring Large Language Models by Model Editing. In The Twelfth International Conference on Learning Representations.

---

> ### Author Response · Authors · 2025-11-22
> **Response to Reviewer 4zDa (Part 2)**
>
> > W2: No effectiveness comparison with a baseline of a single-step backdoor attack.
>
>
> 1. CoTri aims to achieve continuous control across every step of agentic tasks. This fundamental objective distinguishes it from single-step backdoor attacks in terms of application scenarios and control requirements, making a direct performance comparison inappropriate. We have also discussed the specific limitations and performance of single-step backdoors in our response to W1.
>
> 2. The "success" of step 1 denotes only the execution of the initial action (e.g., searching for a copier), not the completion of the full attack. Therefore, this single-point success cannot replace the multi-step design essential for agentic tasks. Our method aims for continuous control at every step of the agent's execution, guiding it along the most direct path to the target. This achieves an unattainable efficiency and effectiveness by single-step backdoors.
>
> 3. The stealthiness of CoTri is not correlated with the number of attack steps. First, because subsequent triggers are from the environment itself, they possess inherent stealth, appearing as natural parts of the task rather than inserted anomalies. Second, our training strategy ensures that the agent reverts to benign behavior whenever the trigger chain is incomplete. The backdoored agent even exhibits enhanced task performance and robustness in noisy environments. This improvement over the baseline model makes the attack highly stealthy, as the agent appears more capable rather than compromised.
>
> >W3: Limited technological contribution in attacking CoT with a backdoor
>
> 1.  The core difference between CoTri and [a] is essential, regarding the task domain:
>     - Prior Work [a] (Li et al.): The proposed "multi-trigger backdoor" is a "multi-conditional constraint at a single time step." Its design goal is to embed multiple triggers (parallel, serial, or mixed) to counter the risk of a backdoor shortcut being detected and pruned. The core application scenario is static tasks (e.g., single image recognition).
>     - CoTri: Our trigger sequence is distributed across multiple time steps." The initial trigger is an inserted textual token, but subsequent triggers are entirely derived from the task environment. This represents a novel design tailored for agentic tasks, differing completely from the "single-node multi-condition" logic of the cited work.
> 2. For technical contribution, we innovatively constructed a poisoning dataset to explicitly model environmental randomness.
>     - This process not only achieves a reliable backdoor but also enhances the agent's robustness: when a trigger in a specific step is missed, the agent automatically reverts to the benign task performance; when the full trigger sequence appears completely and correctly, the backdoor stably activates, ensuring high attack efficacy.
>     - We further demonstrate it through specific data ratios (mixing valid and invalid examples) tailored to the different requirements across various agent models and input modalities.
>
>
> >Q1: What is the benefit of employing the multiple-step backdoor attack instead of a single-step one, and how does the benefit demonstrate in their evaluation?
>
> Please refer to our responses in W1, W2 and W3.

---

### Note · Authors · 2026-01-05

I have read and agree with the venue's withdrawal policy on behalf of myself and my co-authors.